# Towards a General Transfer Approach for Policy-Value Networks

**Dennis J.N.J. Soemers**\*  *dennis.soemers@maastrichtuniversity.nl*
*Department of Advanced Computing Sciences, Maastricht University*

**Vegard Mella**  *vegard.mella@gmail.com*
*Meta AI Research*

**Éric Piette**  *eric.piette@uclouvain.be*
*ICTEAM, UCLouvain*

**Matthew Stephenson**  *matthew.stephenson@flinders.ed.au*
*College of Science and Engineering, Flinders University*

**Cameron Browne**  *cameron.browne@maastrichtuniversity.nl*
*Department of Advanced Computing Sciences, Maastricht University*

**Olivier Teytaud**  *oteytaud@meta.com*
*Meta AI Research*

**Reviewed on OpenReview:** *https://openreview.net/forum?id=vJcTm2v9Ku*

## Abstract

Transferring trained policies and value functions from one task to another, such as one game to another with a different board size, board shape, or more substantial rule changes, is a challenging problem. Popular benchmarks for reinforcement learning (RL), such as Atari games and ProcGen, have limited variety especially in terms of action spaces. Due to a focus on such benchmarks, the development of transfer methods that can also handle changes in action spaces has received relatively little attention. Furthermore, we argue that progress towards more general methods should include benchmarks where new problem instances can be described by domain experts, rather than machine learning experts, using convenient, high-level domain specific languages (DSLs). In addition to enabling end users to more easily describe their problems, user-friendly DSLs also contain relevant task information which can be leveraged to make effective zero-shot transfer plausibly achievable. As an example, we use the Ludii general game system, which includes a highly varied set of over 1000 distinct games described in such a language. We propose a simple baseline approach for transferring fully convolutional policy-value networks, which are used to guide search agents similar to AlphaZero, between any pair of games modelled in this system. Extensive results—including various cases of highly successful zero-shot transfer—are provided for a wide variety of source and target games.

## 1 Introduction

While deep reinforcement learning (RL) (Sutton & Barto, 2018) models have produced many strong and noteworthy results (Mnih et al., 2015; Silver et al., 2016; Brown & Sandholm, 2019; Schrittwieser et al., 2020), they are also frequently associated with limitations in terms of generalisation and transfer (Rusu et al., 2016; Marcus, 2018; Zhang et al., 2018; Justesen et al., 2018; Cobbe et al., 2019; Zhang et al., 2020;

---

\*Work primarily done during a research internship at Meta AI Research.

Zhu et al., 2020). For example, models trained using AlphaGo (Silver et al., 2016) or its successors (Silver et al., 2017; 2018) can perform well in the exact games they are trained on (such as Go on a 19×19 board), but cannot at all play even slight variants (such as Go on a 17×17 board).

Standardised benchmarks and frameworks with collections of environments play an important role in evaluating and advancing deep RL methods. Examples of such frameworks include the Arcade Learning Environment (ALE) (Bellemare et al., 2013; Machado et al., 2018), OpenAI Gym (Brockman et al., 2016), ProcGen (Cobbe et al., 2020), and MiniHack (Samvelyan et al., 2021). Most of these are restricted in terms of extensibility or variety of environments. ALE contains a substantial number of games, but they are all arcade games, and all restricted to a discrete action space of up to 18 elements—most of which are movement directions that are semantically similar between different games. ProcGen was specifically created to benchmark generalisation in RL, but it is restricted to environments with exactly 15-dimensional discrete action spaces, and 64×64×3 RGB state observations. Hence, while it can benchmark generalisation across levels, it cannot benchmark generalisation across state or action representations. OpenAI Gym can in principle be extended to include any environment, but adding new environments consists of programming them from scratch (e.g., in Python), and determining their state and action representations by hand. Except for MiniHack, none of the frameworks listed above include any explicit form of task descriptions (arguably except for their raw source). This means that effective zero-shot transfer can never be reasonably expected to be possible when tasks are made sufficiently varied in these frameworks, in the same way that humans also cannot be expected to continue playing well if a game's rules are changed without informing the human.

GVGAI (Schaul, 2013; Perez-Liebana et al., 2019), Ludii (Browne et al., 2020; Piette et al., 2020), Stratega (Perez-Liebana et al., 2020), Griddly (Bamford et al., 2021), and MiniHack (Samvelyan et al., 2021) are examples of frameworks that use relatively high-level domain-specific languages (DSLs) or game description languages (GDLs) to describe the rules of games and/or their levels. Such high-level languages are substantially easier to use for defining new environments than a general-purpose programming language. Over 1000 games have already been written in Ludii's GDL in a few years' time, of which approximately 10% were contributed by third-party users, including users with backgrounds in (board) game design rather than programming. For research towards more generally applicable artificial intelligence, we consider that models should support highly extensible and varied problem sets, where problem instances are defined by end users (such as domain experts) using such convenient languages, rather than machine learning experts or software engineers. While the frameworks listed as examples above are all for games, similar DSLs can also be used to describe different families of problems, such as staff scheduling problems (Schafhauser, 2010) or algorithm design (Sironi & Winands, 2021). High-level DSLs have also been shown to facilitate evolutionary generation of interesting new problems (such as new games) (Browne, 2009), which is another interesting property that may benefit the training of generalisable agents (Justesen et al., 2018; Cobbe et al., 2020; Jiang et al., 2021; Parker-Holder et al., 2022).

This paper focuses on transfer of policy-value networks—as trained by AlphaZero-based (Silver et al., 2018) approaches—between games in Ludii. We focus on AlphaZero-based approaches because it is a standard technique that is known to work well in 2-player zero-sum games (Silver et al., 2018; Cazenave et al., 2020), and also provides an interesting challenge for transfer in that it involves both policy and value heads. All of the frameworks with DSLs listed above have a primary focus on a certain category of games. For example, GVGAI focuses on avatar-centric arcade games with action spaces of up to a size of 5, MiniHack focuses on NetHack-like games (Küttler et al., 2020), and Ludii primarily includes board games (but also some deduction puzzles and other abstract games). Of these, Ludii arguably has the greatest variety in domains, with many different board geometries (Browne et al., 2022) and action spaces. This makes it particularly interesting and challenging for transfer in deep RL, and allows for many different types of targeted experiments; it is easy to change board sizes, board shapes, flip win conditions, and so on, and evaluate how such different types of changes affect transfer.

We propose a relatively simple baseline approach for transfer between Ludii games, and extensively evaluate zero-shot transfer as well as transfer with fine-tuning in a wide variety of pairs of source and target games. One of two key ingredients of the transfer approach is the use of fully convolutional networks with global pooling for their ability to adjust to changes in the spatial dimensions of a game (i.e., changes in board sizes) (Shelhamer et al., 2017; Wu, 2019; Cazenave et al., 2020). In board games this is also typically associated

with substantial changes in the size of the action space. This has been neglected in related work on transfer for deep RL, which almost exclusively tends to consider transfer between state spaces or representations. We use the AlphaZero-like training algorithms as implemented in Polygames (Cazenave et al., 2020) for training initial models and fine-tuning. The second key ingredient is that we leverage the shared, uniform state and action representations (Piette et al., 2021) that result from describing problems in a DSL or GDL like Ludii's to identify shared semantics in the non-spatial aspects of source and target domains.

Source and target domains are sometimes minor variants of the same game (e.g., with a single change in board size, geometry, or win condition), and sometimes different games altogether with a single shared theme (e.g., many different line-completion games that all have some form of line completion as win condition). In total, we report results for 227 different pairings of source and target domains. We identify particularly successful transfer results—including highly successful zero-shot transfer—when transferring from smaller games (often smaller boards) to larger ones, but also several other cases of successful transfer, and some with negative transfer (Zhang et al., 2020).

## 2 Background

This section provides relevant background information on self-play training, tensor representations, and fully convolutional architectures for transfer, in Polygames and Ludii.

### 2.1 Learning to Play Games in Polygames

Similar to AlphaZero, game-playing agents in Polygames use a combination of Monte-Carlo tree search (MCTS) (Kocsis & Szepesvári, 2006; Coulom, 2007) and deep neural networks (DNNs). Experience for training is generated from self-play games between MCTS agents that are guided by the DNN. Given a tensor representation of an input state $s$, the DNN outputs a value estimate $V(s)$ of the value of that state, as well as a discrete probability distribution $\mathbf{P}(s) = [P(s, a_1), P(s, a_2), \ldots, P(s, a_n)]$ over an action space of $n$ distinct actions. Both of these outputs are used to guide the MCTS-based tree search. The outcomes (typically losses, draws, or wins) of self-play games are used as training targets for the value head (which produces $V(s)$ outputs), and the distribution of visit counts to children of the root node by the tree search process is used as a training target for the policy head (which produces $\mathbf{P}(s)$ outputs).

For board games, input states $s$ are customarily represented as three-dimensional tensors of shape $(C, H, W)$, where $C$ denotes a number of channels, $H$ denotes the height of a 2D playable area (e.g., a game board), and $W$ denotes the width. The latter two are interpreted as spatial dimensions by convolutional neural networks. It is typically assumed that the complete action space can be feasibly enumerated in advance, which means that the shape of $\mathbf{P}(s)$ output tensors can be constructed such that every possibly distinct action $a$ has a unique, matching scalar $P(s, a)$ for any possible state $s$ in the policy head. A DNN first produces *logits* $L(s, a)$ for all actions $a$, which are transformed into probabilities using a softmax after masking out any actions that are illegal in $s$.

In some general game systems, it can be impossible to guarantee that multiple different actions will never share a single output in the policy head (Soemers et al., 2022). We say that distinct actions are *aliased* if they are represented by a single, shared position in the policy head's output tensor. In Polygames, the MCTS visit counts of aliased actions are summed up to produce a single shared training target for the corresponding position in the policy head. In the denominator of the softmax, we only sum over the distinct logits that correspond to legal actions (i.e., logits for aliased actions are counted only once). All aliased actions $a$ receive the same prior probability $P(s, a)$ to bias the tree search—because the DNN cannot distinguish between them—but the tree search itself can still distinguish between them.

### 2.2 The Ludii General Game System

Ludii (Browne et al., 2020; Piette et al., 2020) is a general game system with over 1000 built-in games, many of which support multiple variants with different board sizes, board shapes, rulesets, etc. It provides suitable object-oriented state and action representations for any game described in its game description

language, and these can be converted into tensor representations in a consistent manner with no need for additional game-specific engineering effort (Soemers et al., 2022). All games in Ludii are modelled as having one or more "containers," which can be viewed as areas with spatial semantics (such as boards) that contain relevant elements of game states and positions that are affected by actions. This means that all games in Ludii are compatible with fully convolutional architectures from an engineering point of view, although it is possible to describe games where spatial semantics have little meaning and the inductive biases encoded in convolutional layers are ineffective.

### 2.3 Transfer of Policy-Value Networks: the Case for Fully Convolutional Architectures

AlphaZero-like training approaches (Silver et al., 2018) typically require training from scratch before any new individual (variant of a) game can be played. Transfer learning (Taylor & Stone, 2009; Lazaric, 2012; Zhu et al., 2020) may allow for significant savings in computation costs by transferring trained parameters from a *source domain* (i.e., a game that we train on first) to one or more *target domains* (i.e., variants of the source game or different games altogether). We focus on transferring complete policy-value networks—with policy as well as value heads—between a large variety of source and target domains as provided by Ludii. Crucially, the transfer of trained parameters for a network with a policy head requires the ability to construct a mapping between action spaces of source and target domains. This is in contrast to approaches that only transfer value functions, which only require the ability to create mappings between state spaces.

As per Subsection 2.1, we assume that game states $s$ are represented as tensors of shape $(C_{state}, H, W)$, and that the action space can be represented by a shape $(C_{action}, H, W)$—we may think of actions as being represented by an action channel, and coordinates in the 2D space of $H$ rows and $W$ columns. A network's policy head therefore also has a shape of $(C_{action}, H, W)$. We focus on 2-player zero-sum games, which means that a single scalar suffices as output for the value head. Note that different games may have different numbers of channels and different values for $H$ and $W$. It is common to use architectures that first process input states $s$ using convolutional layers, but at the end use one or more non-convolutional layers (e.g., fully-connected layers) preceding the outputs. This leads to networks that cannot handle changes in the spatial dimensions (i.e., changes in $H$ or $W$), and have no inductive biases that leverage any spatial structure that may be present in the action representation. Fully convolutional architectures with global pooling, such as those implemented in Polygames, can address both of those concerns (Wu, 2019). In particular the ability to handle changes in spatial dimensions is crucial for transfer between distinct games or game variants, which is why we restrict our attention to these architectures in the majority of this paper. An example of such an architecture is depicted in Appendix A.

Formally, the goal of transfer is as follows. Let $\mathcal{S}$ denote a source task (i.e., a game we train in and transfer from), with a state space $S^{\mathcal{S}}$ and an action space $A^{\mathcal{S}}$. A trained policy-value network can produce value outputs $V(s)$, and policy outputs $\pi(s, a)$, for (tensor representations of) input states $s \in S^{\mathcal{S}}$ and actions $a \in A^{\mathcal{S}}$. Let $\mathcal{T}$ denote a target task (i.e., a game we transfer to and evaluate performance of agents in), with (potentially) different state and action spaces $S^{\mathcal{T}}$ and $A^{\mathcal{T}}$. Then, the goal of transfer is to use a network trained in $\mathcal{S}$ to initialise the weights of a new neural network, such that the new network can directly perform well and/or speed up learning in $\mathcal{T}$.

## 3 Transferring Parameters Between Games

The fully convolutional architectures as described in the previous section allow for DNNs trained in a source task $\mathcal{S}$ to be directly used in any target task $\mathcal{T}$ if they have different values only for one or both of the spatial dimensions $H$ and $W$. However, it is also possible that different tasks have different numbers of channels $C_{state}$ or $C_{action}$ for state or action representations, or significantly different semantics for some of those channels. We propose a simple baseline approach for transferring the weights of fully convolutional policy-value networks in this setting.

The key insight that we use is that, when a wide variety of problems are all described in a single, high-level DSL—as is the case with over 1000 distinct games in Ludii—the framework can generally represent their states and actions in a uniform, consistent manner, where the different variables used internally closely

correspond to the semantically meaningful high-level keyword that are present in the DSL (Piette et al., 2021; Soemers et al., 2022). Based on these consistent representations across the wide variety of problems that can be modelled in the same language, we can automatically identify variables of state or action representations that are semantically similar in terms of the raw data that they hold. In the case of states, these variables correspond to channels of the input tensors. In the case of actions, these variables correspond to channels of the policy output tensors. For example, an input plane that encodes which position was the target of the most recent move in a source game is said to be semantically "equivalent" to another input plane that encodes the target position of the most recent move for a target game. Note that this idea of equivalence is based on the data that is encoded, but not necessarily on how it affects strategies or optimal policies in a game. For example, planes that encode which position was the target of the most recent move for two different games are considered semantically equivalent in terms of the raw data that they encode, but they may have different implications in terms of the optimal strategies for different games. It may be possible to automatically learn how to account for such differences (Kuhlmann & Stone, 2007; Bou Ammar, 2013; Bou Ammar et al., 2014), but we consider this to be outside the scope of this paper.

Given equivalence relations between state and action channels for $\mathcal{S}$ and $\mathcal{T}$, we can copy parameters from a network trained in $\mathcal{S}$ to play $\mathcal{T}$. Without prior knowledge of the specific games under consideration, there is no explicit guidance on how to transfer other than the equivalence relations. Therefore, to build a rigorous notion of what would constitute "correct" transfer, we start out with the assumption that $\mathcal{S}$ and $\mathcal{T}$ *are exactly the same game, but with different state or action tensor representations*; some information may have been added, removed, or shuffled around. Under this assumption, we can think of correct transfer as any approach that ensures that, as much as possible, equivalent input states lead to equivalent outputs. In practice, we can then relax the assumption of $\mathcal{S}$ and $\mathcal{T}$ being identical games, but transfer parameters in the same way. The extent to which this may be expected to be effective depends on (i) the extent to which similar data (e.g., a channel distinguishing between empty and non-empty positions) has similar implications for optimal policies or value functions in a different (representation of a) game, as well as (ii) the extent to which the different domains contain different types of information that cannot be objectively mapped to each other. For example, when $\mathcal{S}$ and $\mathcal{T}$ have different types of pieces (e.g., *Shogi* has the *Kyosha* piece type, which *Minishogi* does not), channels encoding the presence of such pieces are heuristically mapped based on tree edit distances (Zhang & Shasha, 1989) between the trees of code describing the movement rules of those pieces in Ludii's GDL.

Transferring parameters such that identical inputs produce, where possible, identical outputs, as described above, can be done by copying, reordering, duplicating, or removing channels of the first and final convolutional layers according to the equivalence relations. This may sometimes only be approximately possible, because differences in representation can mean that $\mathcal{T}$ contains less information than $\mathcal{S}$; either representation may have only partial information in state tensors, or some degree of action aliasing. Parameters of layers other than the first and final convolutional layers can be directly copied. The remainder of this section provides technical details on the transfer of parameters between $\mathcal{S}$ and $\mathcal{T}$ (building on the assumption of those domains being the same game), based on equivalence relations between channels.[1] The identification of equivalence relations between channels is, assuming that $\mathcal{S}$ and $\mathcal{T}$ are described in Ludii's GDL, fully automated. More extensive details on the transfer of parameters, as well as on the identification of equivalence relations in Ludii, are provided in Appendix B. Appendix D provides a full example of how state channels are mapped between the games of *Minishogi* and *Shogi*.

### 3.1 Transferring State Channel Parameters

Let $s^{\mathcal{S}}$ denote the tensor representation for any arbitrary state in $\mathcal{S}$, with shape $(C_{state}^{\mathcal{S}}, H, W)$. Let $s^{\mathcal{T}}$ denote the tensor representation, of shape $(C_{state}^{\mathcal{T}}, H, W)$, for the same state represented in $\mathcal{T}$—this must exist by the assumption that $\mathcal{S}$ and $\mathcal{T}$ are the same game, modelled in different ways. Let $h_1^{\mathcal{S}}(s^{\mathcal{S}})$ denote the hidden representation obtained by the application of the first convolutional operation on $s^{\mathcal{S}}$, in a network trained on $\mathcal{S}$. For brevity we focus on the case used in our experiments, but most if not all of these assumptions can likely be relaxed; an `nn.Conv2d` layer as implemented in PyTorch (Paszke et al., 2019), with 3×3 filters, a stride and dilation of 1, and a padding of 1 (such that spatial dimensions do not change).

---

[1]Source code used to transfer weights: `https://github.com/DennisSoemers/Transfer-DNNs-Ludii-Polygames`.

Let $\Theta^{\mathcal{S}}$ denote this layer's tensor of parameters trained in $\mathcal{S}$, of shape $(k_{out}, C_{state}^{\mathcal{S}}, 3, 3)$, and $B^{\mathcal{S}}$—of shape $(k_{out})$—the corresponding bias. The shape of $h_1^{\mathcal{S}}$ is $(k_{out}, H, W)$. Similarly, let $h_1^{\mathcal{T}}(s^{\mathcal{T}})$ denote the first hidden representation in the network after transfer, for the matching state $s^{\mathcal{T}}$ in the new representation of $\mathcal{T}$, with weight and bias tensors $\Theta^{\mathcal{T}}$ and $B^{\mathcal{T}}$.

Under the assumption that $\mathcal{S}$ and $\mathcal{T}$ are identical, we aim to ensure that $h_1^{\mathcal{S}}(s^{\mathcal{S}}) = h_1^{\mathcal{T}}(s^{\mathcal{T}})$. This means that the first convolutional layer will handle any changes in the state representation, and the remainder of the network can be transferred in its entirety and behave as it learned to do in $\mathcal{S}$. $B^{\mathcal{T}}$ is simply initialised by copying $B^{\mathcal{S}}$. Let $i \cong j$ denote that channel $i$ in $\mathcal{S}$ has been determined to be semantically equivalent to channel $j$ in $\mathcal{T}$. For any channel $j$ in $\mathcal{T}$, if there exists a channel $i$ in $\mathcal{S}$ such that $i \cong j$, we initialise $\Theta^{\mathcal{T}}(k, j, \cdot, \cdot) := \Theta^{\mathcal{S}}(k, i, \cdot, \cdot)$ for all $k$. If there is no such channel $i$, we initialise these parameters as new untrained parameters (or initialise them to 0 for zero-shot evaluations).

If $\mathcal{T}$ contains channels $j$ such that there are no equivalent channels $i$ in $\mathcal{S}$ (i.e., $\{i \mid i \cong j\} = \varnothing$), we have no transfer to the $\Theta^{\mathcal{T}}(k, j, \cdot, \cdot)$ parameters. This can be the case if $\mathcal{T}$ involves new data for which there was no equivalent in the representation of $\mathcal{S}$, which means that there was also no opportunity to learn about this data in $\mathcal{S}$. If $\mathcal{S}$ contained channels $i$ such that there are no equivalent channels $j$ in $\mathcal{T}$, i.e. $\nexists j (i \cong j)$, we have no transfer from the $\Theta^{\mathcal{S}}(k, i, \cdot, \cdot)$ parameters. This can be the case if $\mathcal{S}$ involved data that is no longer relevant or accessible in $\mathcal{T}$. Not using them for transfer is equivalent to pretending that these channels are still present, but always filled with 0 values.

If $\mathcal{S}$ contained a channel $i$ such that there are multiple equivalent channels $j$ in $\mathcal{T}$, i.e. $|\{j \mid i \cong j\}| > 1$, we copy a single set of parameters multiple times. This can be the case if $\mathcal{T}$ uses multiple channels to encode data that was encoded by just a single channel in $\mathcal{S}$ (possibly with loss of information). In the case of Ludii's tensor representations, this only happens when transferring channels representing the presence of piece types from a game $\mathcal{S}$ with fewer types of pieces, to a game $\mathcal{T}$ with more distinct piece types. In the majority of games in Ludii, such channels are "mutually exclusive" in the sense that, if a position contains a 1 entry in one of these channels, all other channels in the set are guaranteed to have a 0 entry in the same position. This is because most games only allow a single piece to occupy any given position of the board. This means that copying the same parameters multiple times can still be viewed as "correct"; for any given position in a state, they are guaranteed to be multiplied by a non-zero value at most once. The only exceptions are games $\mathcal{T}$ that allow for multiple pieces of distinct types to be stacked on top of each other in a single position, but these games are rare and not included in any of the experiments described in this paper.

Cases where $\mathcal{T}$ contains a single channel $j$ such that there are multiple equivalent channels $i$ in $\mathcal{S}$ (i.e., $|\{i \mid i \cong j\}| > 1$) never occur when channels are mapped in Ludii as described in Appendix B.

### 3.2 Transferring Action Channel Parameters

Let $s^{\mathcal{S}}$ and $a^{\mathcal{S}}$ denote any arbitrary state and action in $\mathcal{S}$, such that $a^{\mathcal{S}}$ is legal in $s^{\mathcal{S}}$, and let $s^{\mathcal{T}}$ and $a^{\mathcal{T}}$ denote the corresponding representations in $\mathcal{T}$. We assume that the state representations have been made equivalent through transfer of parameters for the first convolutional layer, as described in Subsection 3.1. Let $h_n(s^{\mathcal{S}})$ be the hidden representation that, in a fully convolutional architecture trained in $\mathcal{S}$, is transformed into a tensor $L(h_n(s^{\mathcal{S}}))^{\mathcal{S}}$ of shape $(C_{action}^{\mathcal{S}}, H, W)$ of logits. Similarly, let $L(h_n(s^{\mathcal{T}}))^{\mathcal{T}}$ of shape $(C_{action}^{\mathcal{T}}, H, W)$ denote such a tensor of logits in $\mathcal{T}$. By assumption, we have that $h_n(s^{\mathcal{S}}) = h_n(s^{\mathcal{T}})$.

If the action representations in $\mathcal{S}$ and $\mathcal{T}$ are equally powerful in their ability to distinguish actions, we can define a notion of correct transfer of parameters by requiring the transfer to ensure that $L(h_n(s^{\mathcal{S}}))^{\mathcal{S}} = L(h_n(s^{\mathcal{T}}))^{\mathcal{T}}$ after accounting for any shuffling of channel indices. If we have one-to-one mappings for all action channels, this can be easily achieved by copying parameters of the final convolutional operation in a similar way as for state channels (see Subsection 3.1).

If the action representation of $\mathcal{T}$ can distinguish actions that cannot be distinguished in $\mathcal{S}$, we have a reduction in move aliasing. This happens when transferring from placement games to movement games. Since these actions were treated as identical in $\mathcal{S}$, we give them equal probabilities in $\mathcal{T}$ by mapping a single source channel to multiple target channels, and copying trained parameters accordingly. If $\mathcal{S}$ could distinguish actions from each other that $\mathcal{T}$ cannot distinguish, we have an increase in move aliasing (loss of

information). This case is handled conservatively by only allowing transfer from a single source channel—the one that is arguably the "most similar"—and discarding all other parameters.

## 4 Experiments

This section discusses experiments[2] used to evaluate the performance of fully convolutional architectures, as well as transfer learning between variants of games and between distinct games. We used the training code from Polygames. For transfer learning experiments, we used games as implemented in Ludii v1.1.6. Every used game is a zero-sum game for two players. All games may be considered sparse-reward problems in the sense that they only have potential non-zero rewards when terminal game states are reached, and no reward shaping is used. Some of the games, such as *Breakthrough* and *Hex*, naturally progress towards terminal game states with non-zero rewards regardless of player strength (even under random play), whereas others also have terminal game states with outcomes equal to 0 for both players (e.g., *Diagonal Hex*). Appendix E provides details on hyperparameter values used for training throughout all experiments.

### 4.1 Fully Convolutional Architectures

Our approach for transfer requires the use of fully convolutional architectures with global pooling to facilitate transfer between domains with differences in spatial aspects (e.g., different board sizes). The goal of our first experiment is to ensure that the choice for this type of architecture—as opposed to the more common architectures with fully-connected layers prior to the outputs (Silver et al., 2018)—does not come at a substantial cost in baseline performance. We selected a variety of board games as implemented in Polygames, and trained networks of various sizes and architectures, using 24 hours on 8 GPUs and 80 CPU cores per model. Models of various sizes—measured by the number of trainable parameters—have been constructed by randomly drawing choices for hyperparameters such as the number of layers, blocks, and channels for hidden layers. After training, we evaluated the performance of every model by recording the win percentage of an MCTS agent using 40 iterations per move with the model, versus a standard untrained UCT (Browne et al., 2012) agent with 800 iterations per move. The untrained UCT backs up the average outcome of ten random rollouts from the node it traverses to in each iteration. These win percentages are depicted in Figure 1. In the majority of cases, `ResConvConvLogitPoolModel`—a fully convolutional model with global pooling—is among the strongest architectures. Fully convolutional models generally outperform ones with dense layers, and models with global pooling generally outperform those without global pooling. This suggests that using such architectures can be beneficial in and of itself, and their use to facilitate transfer learning does not lead to a sacrifice in baseline performance. Therefore, all transfer learning experiments discussed below used the `ResConvConvLogitPoolModel` architecture from Polygames. All models were trained for 20 hours on 8 GPUs and 80 CPU cores, using 1 server for training and 7 clients for the generation of self-play games.

### 4.2 Transfer Between Game Variants

We selected a set of nine different board games, as implemented in Ludii, and for each of them consider a few different variants. The smallest number of variants for a single game is 2, and the largest number of variants for a single game is 6. In most cases, the different variants are simply different board sizes. For example, we consider *Gomoku* played on 9×9, 13×13, 15×15, and 19×19 boards as four different variants of *Gomoku*. We also include some cases where board shapes change (e.g., *Breakthrough* played on square boards as well as hexagonal boards), "small" changes in rules (e.g., *Broken Line* played with a goal line length of 3, 4, 5, or 6), and "large" changes in rules (i.e., *Hex* with the standard win condition and *Misère Hex* with an inverted win condition). The set of games is highly varied in aspects such as sizes of state and action spaces, board geometries (square boards and three different board shapes with hexagonal cells), types of win and loss conditions (based on completing lines of pieces, connecting chains of pieces, reaching specific areas of the board, or blocking the opponent from having any legal moves), move mechanisms (some games involve placing pieces anywhere on the board, others involve small steps or longer-range slides of pieces, or hopping over opposing pieces), mechanisms for removing pieces (in some games pieces are never removed,

---

[2]Source code: `https://github.com/DennisSoemers/Transfer-DNNs-Ludii-Polygames`.

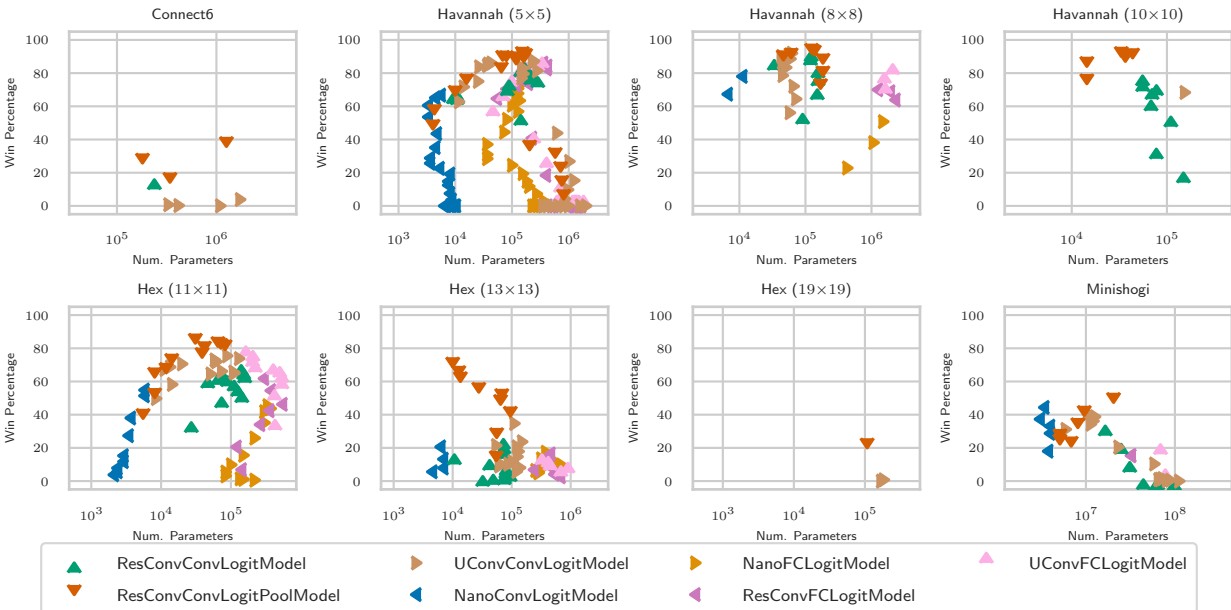

Figure 1: Win percentages of trained MCTS with 40 iterations/move vs. UCT with 800 iterations/move. `Nano`: shallow architectures. `ConvConv`: deep fully convolutional. `ConvFC`: fully connected layers after convolutional ones. `Pool`: adding global pooling; `U`: adding U-connections (Ronneberger et al., 2015). Deep is better than shallow, U-nets are slightly better than their classical counterparts, and deep nets are improved by using (i) fully convolutional policy heads (ii) global pooling.

others involve capturing pieces by landing on them, hopping over them, or surrounding them), and so on. Further details on all the games and game variants used are provided in Appendix F.

We trained a separate model for every variant of each of these games, and transferred models from all variants to all other variants within each game. Only one model (with one random seed) is trained per game, but AlphaZero-based training processes is generally observed to be highly reproducible (Silver et al., 2018). We evaluate zero-shot transfer performance for a source domain $\mathcal{S}$ and target domain $\mathcal{T}$ by reporting the win percentage of the model trained in $\mathcal{S}$ against the model that was trained in $\mathcal{T}$, over 300 evaluation games per $(\mathcal{S}, \mathcal{T})$ pair running in $\mathcal{T}$. In each set of 300 evaluation games, each agent plays as the first player in 150 games, and as the second player in the other 150 games.

Figures 2(a) and 2(b) depict scatterplots of these zero-shot transfer evaluations for cases where $\mathcal{S}$ has a larger board size than $\mathcal{T}$, and where $\mathcal{S}$ has a smaller board size than $\mathcal{T}$, respectively. Win percentages of the transferred model against the baseline model are on the $y$-axis, and the ratio of the number of training epochs of the source model to the number of epochs of the target model is on the $x$-axis. Models trained on larger board sizes tend to have lower numbers of epochs for three reasons; (i) the neural network passes are more expensive, (ii) the game logic in Ludii is more expensive, and (iii) episodes often tend to last for a higher number of turns when played on larger boards. Hence, points in Figure 2(a) have a ratio $\leq 1.0$, and points in Figure 2(b) have a ratio $\geq 1.0$. Detailed tables with individual results for every $(\mathcal{S}, \mathcal{T})$ pair are listed in Appendix G.

For zero-shot transfer of a model trained on a large board to a small board (Figure 2(a)), win percentages tend to be below 50%, but frequently still above 0%; training on a larger board than the one we intend to play on does not outperform simply training on the correct board directly, but often still produces a capable model that can win a non-trivial number of games. There is a meaningful level of transfer in this direction, but (as would be expected) it is still better to use the same amount of computation time directly on the target domain. Transferring a model from a small board to a large board (Figure 2(b)) frequently leads to win percentages above 50%, even reaching up to 100%, against models trained directly in $\mathcal{T}$. Results above

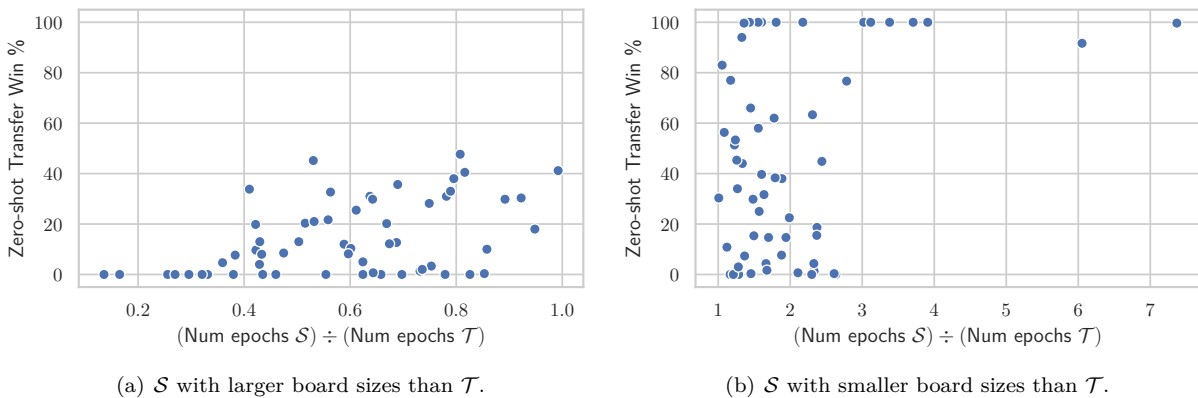

(a) $\mathcal{S}$ with larger board sizes than $\mathcal{T}$.

(b) $\mathcal{S}$ with smaller board sizes than $\mathcal{T}$.

Figure 2: Zero-shot win percentages of models trained on $\mathcal{S}$, against models trained on $\mathcal{T}$—evaluated on $\mathcal{T}$. For zero-shot transfer, any win percentage greater than 0% indicates some level of successful transfer, and results greater than 50% suggest training on $\mathcal{S}$ is better than training on $\mathcal{T}$—even for a subsequent evaluation on $\mathcal{T}$. "Num epochs $\mathcal{S}$" is the number of training epochs that fit within the training time budget for the model trained on $\mathcal{S}$ (the model being evaluated). "Num epochs $\mathcal{T}$" is the number of training epochs for the model trained on $\mathcal{T}$ (the opponent that we evaluate against).

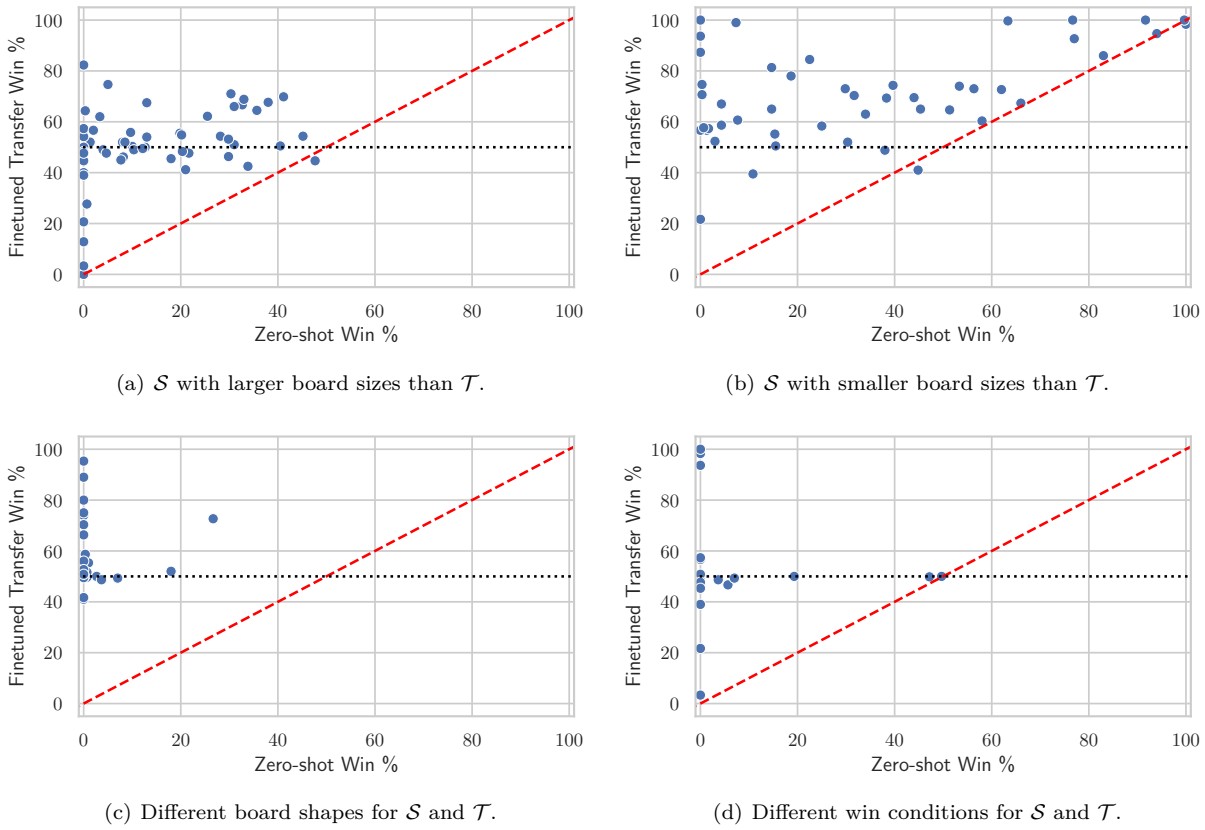

(a) $\mathcal{S}$ with larger board sizes than $\mathcal{T}$.

(b) $\mathcal{S}$ with smaller board sizes than $\mathcal{T}$.

(c) Different board shapes for $\mathcal{S}$ and $\mathcal{T}$.

(d) Different win conditions for $\mathcal{S}$ and $\mathcal{T}$.

Figure 3: Win percentages of models trained on $\mathcal{S}$ and subsequently fine-tuned on $\mathcal{T}$, against models trained only on $\mathcal{T}$—evaluated on $\mathcal{T}$. Results above 50% (the horizontal, black line) may be interpreted as beneficial transfer—assuming that pre-training on $\mathcal{S}$ is "for free", without depleting any budget—whereas results below 50% indicate negative transfer. Results below the red $y = x$ line indicate that finetuning degraded performance relative to the zero-shot transfer performance.

50% are particularly interesting, since they suggest that computation time is spent more wisely by training on different (smaller) boards than the target domain, even in zero-shot settings.

Zero-shot transfer between variants with larger differences, such as changes in board shapes or win conditions, only leads to win percentages significantly above 0% in a few cases. For example, there is some meaningful zero-shot transfer with win percentages of 26.67% and 18.00% for transfer between two different board shapes in *Pentalath*. Figure 6 in the supplementary material illustrates the different shapes of these boards.

For every model transferred from a domain $\mathcal{S}$ to a domain $\mathcal{T}$ as described above, we train it under identical conditions as the initial training runs—for an additional 20 hours—to evaluate the effect of using transfer for initialisation of the network. Figure 3 depicts scatterplots of win percentages for four distinct cases: transfer to variants (a) with smaller board sizes, (b) with larger board sizes, (c) with different board shapes, and (d) with different win conditions. Detailed tables are provided in Appendix H. In these plots, we place the win percentages that were obtained by zero-shot transfer between the same source-target pairings on the $x$-axis. Variants of these plots more similar to Figure 2, with ratios of training epochs on the $x$-axis, are available in Figure 8 of the supplementary material. There are many cases of win percentages close to 50%, which can be interpreted as cases where transfer neither helped nor hurt final performance, and many cases with higher win percentages—which can be interpreted as cases where transfer increased final performance in comparison to training from scratch. We observe a small number of cases, especially when $\mathcal{S}$ has a larger board than $\mathcal{T}$ or has different win conditions, in which there is clear negative transfer (Zhang et al., 2020) and the final performance is still closer to 0% even after fine-tuning on $\mathcal{T}$.

When there are substantial differences between source and target domains, a likely possible cause of any negative transfer (even after fine-tuning) is that a poor initial policy may be detrimental to exploration and the quality of collected experience, which in turn may prevent the training process from escaping the poor initialisation. Another possible cause of negative transfer (or lack of successful transfer) is loss of plasticity (Berariu et al., 2021; Dohare et al., 2021; Abbas et al., 2023; Lyle et al., 2023). The most drastic cases of negative transfer are a 0.00% winrate after transfer with fine-tuning from *Diagonal Hex* (19×19) to (11×11), and a 3.33% winrate for transfer from *Hex* (19×19) to *Misère Hex* (11×11). Interestingly, these are both cases of transfer from connection games on 19×19 hexagonal boards to 11×11 hexagonal boards, but transfer from the same source domains to larger as well as smaller target domains than 11×11 worked better. It is not yet clear why this particular scenario is more challenging, but exactly the same pattern in results emerged again in the repeated runs for ablations described in Subsection 4.4. This observation makes it unlikely that these results are simply statistical anomalies. There are very few cases where playing strength after finetuning is lower than playing strength after zero-shot transfer (results below the red $y = x$ line). Hence, while we do sometimes observe negative transfer in the sense of win percentages lower than 50% after finetuning, these can generally be attributed to the poor initial state that the initial transfer places them in. Performance rarely continues to degrade with finetuning after transfer, and never by a substantial margin.

The experiments described above include two core settings; (i) zero-shot transfer, evaluating training for 20 hours on $\mathcal{S}$ and 0 on $\mathcal{T}$, against 20 hours on $\mathcal{T}$, and (ii) fine-tuning, evaluating training for 20 hours on $\mathcal{S}$ and 20 on $\mathcal{T}$, against just 20 hours on $\mathcal{T}$. A third setting that could be considered would be a variant of the fine-tuning setting, against a baseline with 40 hours of training on $\mathcal{T}$. However, in such a setting, there is no clear threshold that objectively separates successful transfer from unsuccessful transfer. Therefore, we focus our efforts on the two settings discussed above, which more readily allow for objective conclusions.

## 4.3 Transfer Between Different Games

Finally, we selected four sets of games that each share a single overarching theme, and within each set carried out similar experiments as described above—this time transferring models between distinct games, rather than minor variants of the same game. The first set consists of six different **Line Completion Games**; in each of these, the goal is to create a line of $n$ pieces, but games differ in values of $n$, board sizes and shapes, move rules, loss conditions, etc. We evaluate transfer from each of those games, to each of these games. The second set consists of four **Shogi Games**: *Hasami Shogi*, *Kyoto Shogi*, *Minishogi*, and *Shogi*, and evaluate transfer from and to each of them. The third set evaluates transfer from each of four variants of *Broken Line*, to each of the six line completion games. *Broken Line* is a custom-made line completion game where

only diagonal lines count towards the win condition, whereas the standard line completion games allow for orthogonal lines. The fourth set evaluates transfer from each of five variants of *Diagonal Hex*, to each of six variants of *Hex*. *Diagonal Hex* only considers diagonal connections for its win condition, whereas the *Hex* variants only consider orthogonal connections. Appendix F provides more details on all the games and variants.

In most cases, zero-shot win percentages for models transferred between distinct games are close to 0%. We observe some success with win percentages over 30% for transfer from several line completion games to *Connect 6*, win percentages between 20% and 50% for transfer from three different Shogi games to *Hasami Shogi*, and a win percentage of 97% for transfer from *Minishogi* to *Shogi*. The last result is particularly significant, since it suggests that, when the ultimate goal is to train a *Shogi* agent, computation time is spent more efficiently training purely on the distinct game of *Minishogi* (at least for the first 20 hours). Appendix I contains more detailed results.

Figure 4 depicts win percentages for transferred models after they received 20 hours of fine-tuning time on $\mathcal{T}$. We observe many cases of succesful transfer between various line-completion games (including *Broken Line*), as well as several Shogi games. Most notably for various cases of transfer from *Diagonal Hex* to *Hex*, there is a high degree of negative transfer, with win percentages far below 50% even after fine-tuning. It may be that the differences in connectivity rules are too big to allow for consistently successful transfer. It may also be due to large differences in the distributions of outcomes, resulting in mismatches for the value head; ties are common in some variants of *Diagonal Hex*, but impossible in *Hex*.

### 4.4 Variations on Fine-tuning Experiments

For all fine-tuning experiments described throughout Subsections 4.2 and 4.3, we considered two additional variations on the experiments. The first is one where, after transfer but before fine-tuning, all the weights in the final layers are reinitialised. This can reduce the risk of remaining stuck in a poor local minimum in cases where transferred models perform poorly in the target domain, and may mitigate loss of plasticity (Berariu et al., 2021). The second variation is one where, after transfer, randomly selected channels from hidden layers are removed, or new ones are added, to ensure that the transferred model has exactly as many parameters as the original model trained on $\mathcal{T}$ did. Without this additional step, models trained on distinct domains $\mathcal{S}$ and $\mathcal{T}$ sometimes have different numbers of channels in hidden layers give the default settings of Polygames. By default, it uses twice the number of channels in the game's state tensor representation as the number of channels for hidden layers. Neither of these ablations resulted in any significant changes in results. As a side effect, the results from these ablation studies also suggest that the fine-tuning process is highly reproducible, and not sensitive to random seed. Full results are included in Appendices H and J.

## 5 Related Work

ProcGen (Cobbe et al., 2020) is a framework in which levels are procedurally generated, which was proposed to benchmark generalisation in RL. However, its fixed action space across all supported domains means that it is unsuitable for evaluating transfer with changes in action spaces. Many other popular frameworks, such as ALE (Bellemare et al., 2013; Machado et al., 2018) and GVGAI (Perez-Liebana et al., 2019) are restricted to similarly small action spaces. MiniHack (Samvelyan et al., 2021) was proposed as a benchmark for evaluating several aspects of RL, including transfer in RL. It can be used to model domains with substantially larger action spaces than the other frameworks mentioned above, but is still limited to action spaces of at most 98 actions. This is significantly smaller than many board games supported by Ludii, which can easily have action spaces of many hundreds of elements. The action space of MiniHack is also relatively unstructured, with only a small subset of it being structured in the sense that they correspond to compass directions.

Like the benchmarks themselves, related work on algorithms and experiments for transfer in deep RL tends to be limited in terms of generality and variety of domains. Some work focuses only on transfer based on visual elements of states (Glatt et al., 2016; Mittel et al., 2018; Sobol et al., 2018; Gamrian & Goldberg, 2019; Zhu et al., 2020), which does not generalise beyond environments with pixel-based state representations such as Atari games. A substantial amount of work is furthermore limited to avatar-centric, single-agent domains

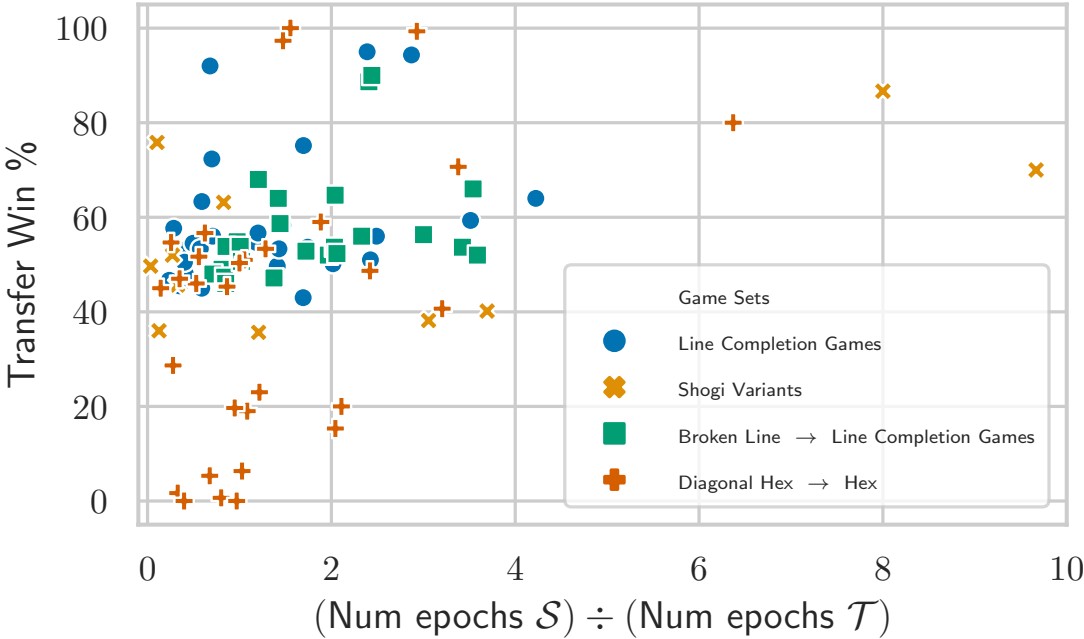

Figure 4: Win percentages of models that were trained in $\mathcal{S}$, transferred to $\mathcal{T}$, and fine-tuned in $\mathcal{T}$, evaluated in $\mathcal{T}$ against models trained directly in $\mathcal{T}$. $\mathcal{S}$ and $\mathcal{T}$ are different games.

with little variety in (often small) action spaces (Rusu et al., 2016; Tessler et al., 2017; Oh et al., 2017; Nichol et al., 2018; Glatt et al., 2020), transfer between domains with identical dynamics (Yang, 2021), or evaluations with only a small number of different games (Asawa et al., 2017). There is some work on learning action embeddings that can aid transfer between domains with different (but related) action spaces (Chen et al., 2021; You et al., 2022), but these approaches rely on experience in the target domain and cannot ever be expected to produce zero-shot results because they do not leverage descriptions of tasks. REvolveR is highly specific robotic control settings, and assumes that it is possible to design continuous interpolations between source and target domains (Liu et al., 2022). Prior to the popularisation of deep learning in the last decade, some work was published on transfer between general games (Banerjee & Stone, 2007; Kuhlmann & Stone, 2007) described in the logic-based GDL from the Stanford Logic Group (Love et al., 2008). However, this work is not directly applicable to deep learning, and not as convenient and extensible as the significantly higher-level more modern GDLs. Gato (Reed et al., 2022) can perform a wide variety of tasks with a single network (i.e., a single set of trained weights). It likely performs some degree of implicit transfer between the various tasks it is trained for. However, it cannot perform zero-shot transfer to new tasks, as considered in this paper. Furthermore, it requires tasks to be described to it in the form of expert trajectories, which is a less general, convenient and practical manner of describing tasks than DSLs.

## 6    Conclusions

In this paper, we argue that research towards more generally applicable artificial intelligence should include models that are applicable to, and can generalise and transfer across, sets of problems that are expressed in high-level domain-specific languages (DSLs). Such sets of problems are easily extensible, and can be created and used by domain experts as end users, rather than programmers or machine learning experts. At the same time, DSLs also provide explicit knowledge of tasks which may facilitate zero-shot transfer. As an example benchmark, we consider Ludii, which contains a highly varied set of over 1000 distinct games expressed in a convenient, high-level game description language. We proposed a relatively simple baseline approach for transferring fully convolutional policy-value networks between pairs of games described in Ludii, and present empirical results for a wide variety of pairings of source and target domains. This includes transferring

models to game variants with modified board sizes, board shapes, inverted win conditions, and different games altogether with more numerous and substantial differences in rules. We provided results for transfer with fine-tuning, but also zero-shot transfer. We identified some cases of negative transfer, but also many with successful transfer. One of the most noteworthy results is that training on a smaller version (e.g., smaller board) of a target game and transferring to a larger version in a zero-shot manner frequently outperforms training for the same amount of time on the actual target domain.

In the implementation of our proposed transfer approach, the identification of related channels between source and target domains is Ludii-specific, but the general design and philosophy behind the approach is generally applicable to any domain where large sets of problems under a common theme are described using a DSL. One avenue for future research would be to incorporate automated, data-driven learning techniques to guide the identification of shared semantics between source and target domains (Kuhlmann & Stone, 2007; Bou Ammar, 2013; Bou Ammar et al., 2014). As a more ambitious and distant future goal, we may consider working towards models that support tasks being described in natural languages, rather than DSLs.

### Acknowledgments

The authors would like to thank Nicolas Usunier for comments on an earlier version of this work. This work was partially supported by the European Research Council as part of the Digital Ludeme Project (ERC Consolidator Grant #771292).

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

## A    Fully Convolutional Policy-Value Network Architecture

A simplified version of the fully convolutional policy-value network architecture considered in this paper is depicted in Figure 5. Note that some details, such as the use of ReLU activations (Nair & Hinton, 2010), batch normalization (Ioffe & Szegedy, 2015), and residual connections (He et al., 2016) have been omitted for brevity. Precise information on details such as numbers of layers, and their sizes, is provided in Appendix E.

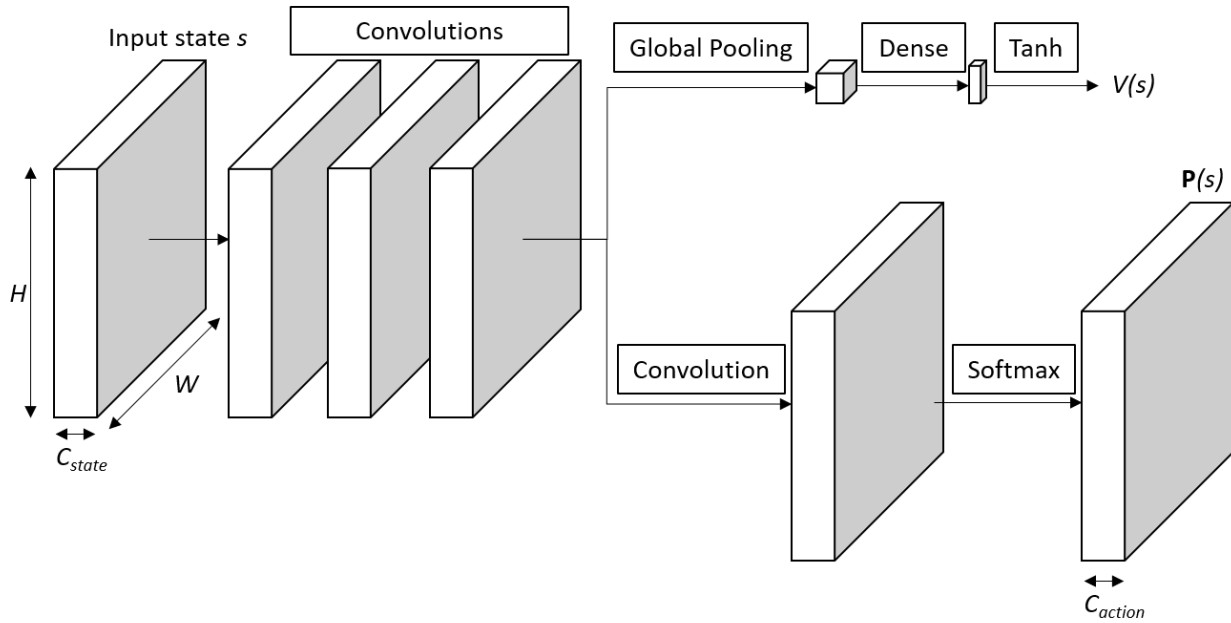

Figure 5: Example of a fully convolutional architecture for game playing. Input states $s$ are provided as tensors of shape $(C_{state}, H, W)$. All convolutions construct hidden representations of shape $(C_{hidden}, H, W)$, where $C_{hidden}$ may differ from $C_{state}$. The policy output $\mathbf{P}(s)$ is a tensor of shape $(C_{action}, H, W)$. The value output $V(s)$ is a scalar. Global pooling is used to reduce the dimensionality for the $V(s)$ output in a manner that can transfer to different board sizes (Wu, 2019; Cazenave et al., 2020).

## B    Details on Transfer of Parameters

In this appendix, we provide extensive technical details on exactly how equivalence relations between channels are determined, and how weights of fully convolutional architectures are transferred accordingly. A higher level overview of this approach is described in Section 3.

We use a relatively simplistic, binary notion of equivalence; a pair of a channel in $\mathcal{S}$ and a channel in $\mathcal{T}$ will either be considered to have identical semantics, or completely different semantics. Furthermore, we only consider the raw data that is encoded by a channel, and do not account for any differences in game rules in this notion of equivalence. For example, the two channels that encode presence of the two players' pieces in many games such as *Hex*, *Go*, *Tic-Tac-Toe*, etc., are considered to have identical semantics.

### B.1    Mapping State Channels

For many channels in Ludii's state tensor representations, semantically equivalent channels are straightforward to identify. For example, a channel that encodes whether player $p$ is the current player, or whether a position was the destination of the previous move in a source domain $\mathcal{S}$, always has an equivalent channel that encodes identical data and can be identified as such in any target domain $\mathcal{T}$. These cases are listed in Appendix C.

The first non-trivial case is that of binary channels which encode, for every pair of spatial coordinates, whether or not the corresponding position exists in a container represented by that channel. For example, *Shogi* in Ludii has three separate containers; a large container for the game board, and two smaller containers representing player "hands," which hold captured pieces. These different containers are assigned different sections of the 2D space, and every container has a binary channel that tells the DNN which positions exist in which containers. In all the built-in games available in Ludii, containers are ordered in a "semantically consistent" manner; the first container is always the main game board, always followed by any player hands, etc. Therefore, we use the straightforward approach of mapping these container-based channels simply in the order in which they appear.

The second case that warrants additional explanation is that of binary channels that encode the presence of pieces; for every distinct piece type that is defined in a game, Ludii creates a binary channel that indicates for every possible position whether or not a piece of that type is placed in that position. Many games (such as *Hex*, *Go*, *Tic-Tac-Toe*, etc.) only have a single piece type per player, and for these we could easily decide that a channel indicating presence of pieces of player $p$ in one game is semantically equivalent to a channel indicating presence of pieces of the same player in another game. For cases with more than a single piece type per player, we partially rely on an unenforced convention that pieces in built-in games of Ludii tend to be named consistently across closely-related games (e.g., similar piece type names are used in many variants of *Chess*). If we find an exact name match for piece types between $\mathcal{S}$ and $\mathcal{T}$, we treat the corresponding channels as semantically equivalent. Otherwise, for any piece type $j$ in $\mathcal{T}$, we loop over all piece types in $\mathcal{S}$, and compute the Zhang-Shasha tree edit distances (Zhang & Shasha, 1989) between the trees of "ludemes" (keywords) that describe the rules for the piece types in Ludii's game description language. Whichever piece type results in the minimum distance to $j$ is assumed to be semantically equivalent to $j$. Appendix D provides a full example of how state channels are mapped between the games of *Minishogi* and *Shogi*.

## B.2 Transferring State Channel Parameters

For the purpose of determining how to transfer parameters that were trained in $\mathcal{S}$ to a network that can play $\mathcal{T}$, we make the assumption that $\mathcal{S}$ and $\mathcal{T}$ *are exactly the same game, but with different state tensor representations*; some state channels may have been added, removed, or shuffled. This assumption lets us build a more rigorous notion of what it means to "correctly" transfer parameters without accounting for differences in rules, optimal strategies, or value functions. In practice, $\mathcal{S}$ and $\mathcal{T}$ can be different games, and we still expect this transfer to be potentially beneficial if the games are sufficiently similar, but the notion of "correct" transfer cannot otherwise be made concrete without domain knowledge of the specific games involved.

Let $s^{\mathcal{S}}$ denote the tensor representation for any arbitrary state in $\mathcal{S}$, with shape $(C_{state}^{\mathcal{S}}, H, W)$. Let $s^{\mathcal{T}}$ denote the tensor representation, of shape $(C_{state}^{\mathcal{T}}, H, W)$, for the same state represented in $\mathcal{T}$—this must exist by the assumption that $\mathcal{S}$ and $\mathcal{T}$ are the same game, modelled in different ways. Let $h_1^{\mathcal{S}}(s^{\mathcal{S}})$ denote the hidden representation obtained by the application of the first convolutional operation on $s^{\mathcal{S}}$, in a network trained on $\mathcal{S}$. For brevity we focus on the case used in our experiments, but most if not all of these assumptions can likely be relaxed; an `nn.Conv2d` layer as implemented in PyTorch (Paszke et al., 2019), with 3×3 filters, a stride and dilation of 1, and a padding of 1 (such that spatial dimensions do not change). Let $\Theta^{\mathcal{S}}$ denote this layer's tensor of parameters trained in $\mathcal{S}$, of shape $(k_{out}, C_{state}^{\mathcal{S}}, 3, 3)$, and $B^{\mathcal{S}}$—of shape $(k_{out})$—the corresponding bias. The shape of $h_1^{\mathcal{S}}$ is $(k_{out}, H, W)$. Similarly, let $h_1^{\mathcal{T}}(s^{\mathcal{T}})$ denote the first hidden representation in the network after transfer, for the matching state $s^{\mathcal{T}}$ in the new representation of $\mathcal{T}$, with weight and bias tensors $\Theta^{\mathcal{T}}$ and $B^{\mathcal{T}}$.

Under the assumption that $\mathcal{S}$ and $\mathcal{T}$ are identical, we can obtain correct transfer by ensuring that $h_1^{\mathcal{S}}(s^{\mathcal{S}}) = h_1^{\mathcal{T}}(s^{\mathcal{T}})$. This means that the first convolutional layer will handle any changes in the state representation, and the remainder of the network can be transferred in its entirety and behave as it learned to do in $\mathcal{S}$. $B^{\mathcal{T}}$ is simply initialised by copying $B^{\mathcal{S}}$. Let $i \cong j$ denote that channel $i$ in $\mathcal{S}$ has been determined to be semantically equivalent to channel $j$ in $\mathcal{T}$. Channels on the left-hand side are source channels, and channels on the right-hand side are target channels. For any channel $j$ in $\mathcal{T}$, if there exists a channel $i$ in $\mathcal{S}$ such that $i \cong j$, we initialise $\Theta^{\mathcal{T}}(k, j, \cdot, \cdot) := \Theta^{\mathcal{S}}(k, i, \cdot, \cdot)$ for all $k$. If there is no such channel $i$, we initialise these

parameters as new untrained parameters (or initialise them to 0 for zero-shot evaluations where subsequent training is unnecessary).

If $\mathcal{T}$ contains channels $j$ such that there are no equivalent channels $i$ in $\mathcal{S}$, i.e. $\{i \mid i \cong j\} = \varnothing$, we have no transfer to the $\Theta^{\mathcal{T}}(k, j, \cdot, \cdot)$ parameters. This can be the case if $\mathcal{T}$ involves new data for which there was no equivalent in the representation of $\mathcal{S}$, which means that there was also no opportunity to learn about this data in $\mathcal{S}$. If $\mathcal{S}$ contained channels $i$ such that there are no equivalent channels $j$ in $\mathcal{T}$, i.e. $\nexists j \, (i \cong j)$, we have no transfer from the $\Theta^{\mathcal{S}}(k, i, \cdot, \cdot)$ parameters. This can be the case if $\mathcal{S}$ involved data that is no longer relevant or accessible in $\mathcal{T}$. Not using them for transfer is equivalent to pretending that these channels are still present, but always filled with 0 values.

If $\mathcal{S}$ contained a channel $i$ such that there are multiple equivalent channels $j$ in $\mathcal{T}$, i.e. $|\{j \mid i \cong j\}| > 1$, we copy a single set of parameters multiple times. This can be the case if $\mathcal{T}$ uses multiple channels to encode data that was encoded by just a single channel in $\mathcal{S}$ (possibly with a loss of information). In the case of Ludii's tensor representations, this only happens when transferring channels representing the presence of piece types from a game $\mathcal{S}$ with fewer types of pieces, to a game $\mathcal{T}$ with more distinct piece types. In the majority of games in Ludii, such channels are "mutually exclusive" in the sense that if a position contains a 1 entry in one of these channels, all other channels in the set are guaranteed to have a 0 entry in the same position. This means that copying the same parameters multiple times can still be viewed as "correct"; for any given position in a state, they are guaranteed to be multiplied by a non-zero value at most once. The only exceptions are games $\mathcal{T}$ that allow for multiple pieces of distinct types to be stacked on top of each other in a single position, but these games are rare and not included in any of the experiments described in this paper.

If $\mathcal{T}$ contains a single channel $j$ such that there are multiple equivalent channels $i$ in $\mathcal{S}$, i.e. $|\{i \mid i \cong j\}| > 1$, there is no clear way to correctly transfer parameters without additional domain knowledge on how the single target channel summarises—likely with a loss of information—multiple source channels. This case never occurs when mapping channels as described in B.1.

## B.3 Mapping Action Channels

Ludii's action tensor representations have three broad categories of channels; a channel for pass moves, a channel for swap moves, and one or more channels for all other moves. Channels for pass or swap moves in one domain can easily be classified as being semantically equivalent only to channels for the same type of moves in another domain.

We refer to games where, in Ludii's internal move representation (Piette et al., 2021), some moves have separate "from" and "to" positions as *movement games* (e.g. *Amazons*, *Chess*, *Shogi*, etc.), and games where all moves only have a "to" position as *placement games* (e.g. *Go*, *Hex*, *Tic-Tac-Toe*, etc.). In placement games, there is only one more channel to encode all moves that are not pass or swap moves. In movement games, there are 49 additional channels, which can distinguish moves based on any differences in $x$ and $y$ coordinates between "from" and "to" positions in $\{\leq -3, -2, -1, 0, 1, 2, \geq 3\}$.

If both $\mathcal{S}$ and $\mathcal{T}$ are placement games, or if both are movement games, we can trivially obtain one-to-one mappings between all move channels. If $\mathcal{S}$ is a movement game, but $\mathcal{T}$ is a placement game, we only treat the source channel that encodes moves with equal "from" and "to" positions (i.e., differences of 0 in both $x$ and $y$ coordinates) as semantically equivalent (in practice, this channel remains unused in the vast majority of movement games, which means that we effectively get no meaningful transfer for moves due to the large discrepancy in movement mechanisms). If $\mathcal{S}$ is a placement game, and $\mathcal{T}$ is a movement game, we treat the sole movement channel from $\mathcal{S}$ as being semantically equivalent to *all* the movement channels in $\mathcal{T}$.

## B.4 Transferring Action Channel Parameters

As in B.2, we make the assumption that $\mathcal{S}$ and $\mathcal{T}$ are identical games, but with *different action tensor representations*, such that we can define a clear notion of correctness for transfer. Let $s^{\mathcal{S}}$ and $a^{\mathcal{S}}$ denote any arbitrary state and action in $\mathcal{S}$, such that $a^{\mathcal{S}}$ is legal in $s^{\mathcal{S}}$, and let $s^{\mathcal{T}}$ and $a^{\mathcal{T}}$ denote the corresponding representations in $\mathcal{T}$. We assume that the state representations have been made equivalent through transfer

of parameters for the first convolutional layer, as described in B.2. Let $h_n(s^\mathcal{S})$ be the hidden representation that, in a fully convolutional architecture trained in $\mathcal{S}$, is transformed into a tensor $L(h_n(s^\mathcal{S}))^\mathcal{S}$ of shape $(C_{action}^\mathcal{S}, H, W)$ of logits. Similarly, let $L(h_n(s^\mathcal{T}))^\mathcal{T}$ of shape $(C_{action}^\mathcal{T}, H, W)$ denote such a tensor of logits in $\mathcal{T}$. By assumption, we have that $h_n(s^\mathcal{S}) = h_n(s^\mathcal{T})$.

If the action representations in $\mathcal{S}$ and $\mathcal{T}$ are equally powerful in their ability to distinguish actions, we can define a notion of correct transfer of parameters by requiring the transfer to ensure that $L(h_n(s^\mathcal{S}))^\mathcal{S} = L(h_n(s^\mathcal{T}))^\mathcal{T}$ after accounting for any shuffling of channel indices. If we have one-to-one mappings for all action channels, this can be easily achieved by copying parameters of the final convolutional operation in a similar way as for state channels (see B.2).

If the action representation of $\mathcal{T}$ can distinguish actions that cannot be distinguished in $\mathcal{S}$, we have a reduction in move aliasing. This happens when transferring from placement games to movement games. Since these actions were treated as identical in $\mathcal{S}$, we give them equal probabilities in $\mathcal{T}$ by mapping a single source channel to multiple target channels, and copying trained parameters accordingly.

When transferring from movement games to placement games, there will be actions that $\mathcal{S}$ could distinguish from each other, but $\mathcal{T}$ cannot. As described previously, we handle this case conservatively by only allowing transfer from a single source channel—the one that is arguably the "most similar"—and discarding all other parameters.

## C  Directly Transferable Ludii State Channels

Most state channels in Ludii's tensor representations (Soemers et al., 2022) are directly transferable between games, in the sense that they encode semantically similar data for any game in which they are present, and can be modelled as channels that always have only values of 0 in any game that they are not present in. These channels are briefly described here:

- A channel encoding the height of a stack of pieces for every position (only present in games that allow for pieces of more than a single piece type to stack on the same position).

- A channel encoding the number of pieces per position (only present in games that allow multiple pieces of the same type to form a pile on the same position).

- Channels encoding the (typically monetary) "amount" value per player.

- Binary channels encoding whether a given player is the current player to move.

- Channels encoding "local state" values per position (for instance used to memorise whether pieces moved to determine the legality of castling in *Chess*).

- A channel encoding whether or not players have swapped roles.

- Channels encoding "from" and "to" positions of the last and second-to-last moves.

## D  Example State Channels for Minishogi and Shogi

Table 1 lists, as an example, all the state channels for the games of *Minishogi* and *Shogi*, and explains which mappings between them are (automatically) determined. Many channels easily mapped in a one-to-one manner, either as described in Appendix C, or based on the names of piece types (many of which are shared between the two games). However, some types of pieces are only present in the larger of the two games (*Shogi*), but absent in *Minishogi*. For example, there are no pieces of the types named "Keima" and "Kyosha" in Minishogi. The piece type named "Hisha" is identified as having the most similar set of rules (according to the tree edit distance Zhang & Shasha (1989) for between the trees of ludemes describing their rules), and these channels are therefore mapped accordingly.

Table 1: State channels for the games of *Minishogi* and *Shogi* in Ludii. When transferring from *Minishogi* to *Shogi*, source state channels from the left column are mapped to target state channels on the same row from the right column. Some channels are repeated multiple times in the left column, because they are used to map to multiple target channels. Repeated occurrences in the left column are printed in italics.

| Minishogi | Shogi |
|---|---|
| 1. Piece Type 1 (Osho1) | 1. Piece Type 1 (Osho1) |
| 2. Piece Type 2 (Osho2) | 2. Piece Type 2 (Osho2) |
| 3. Piece Type 3 (Fuhyo1) | 3. Piece Type 3 (Fuhyo1) |
| 4. Piece Type 4 (Fuhyo2) | 4. Piece Type 4 (Fuhyo2) |
| 5. Piece Type 5 (Ginsho1) | 5. Piece Type 5 (Ginsho1) |
| 6. Piece Type 6 (Ginsho2) | 6. Piece Type 6 (Ginsho2) |
| 7. Piece Type 7 (Hisha1) | 7. Piece Type 7 (Hisha1) |
| 8. Piece Type 8 (Hisha2) | 8. Piece Type 8 (Hisha2) |
| 9. Piece Type 9 (Kakugyo1) | 9. Piece Type 9 (Kakugyo1) |
| 10. Piece Type 10 (Kakugyo2) | 10. Piece Type 10 (Kakugyo2) |
| *7. Piece Type 7 (Hisha1)* | 11. Piece Type 11 (Keima1) |
| *8. Piece Type 8 (Hisha2)* | 12. Piece Type 12 (Keima2) |
| *9. Piece Type 7 (Hisha1)* | 13. Piece Type 13 (Kyosha1) |
| *10. Piece Type 8 (Hisha2)* | 14. Piece Type 14 (Kyosha2) |
| 11. Piece Type 11 (Kinsho1) | 15. Piece Type 15 (Kinsho1) |
| 12. Piece Type 12 (Kinsho2) | 16. Piece Type 16 (Kinsho2) |
| 13. Piece Type 13 (Tokin1) | 17. Piece Type 17 (Tokin1) |
| 14. Piece Type 14 (Tokin2) | 18. Piece Type 18 (Tokin2) |
| 15. Piece Type 15 (Narigin1) | 19. Piece Type 19 (Narigin1) |
| 16. Piece Type 16 (Narigin2) | 20. Piece Type 20 (Narigin2) |
| 17. Piece Type 17 (Ryuo1) | 21. Piece Type 21 (Ryuo1) |
| 18. Piece Type 18 (Ryuo2) | 22. Piece Type 22 (Ryuo2) |
| 19. Piece Type 19 (Ryuma1) | 23. Piece Type 23 (Ryuma1) |
| 20. Piece Type 20 (Ryuma2) | 24. Piece Type 24 (Ryuma2) |
| *11. Piece Type 11 (Kinsho1)* | 25. Piece Type 25 (Narikei1) |
| *12. Piece Type 12 (Kinsho2)* | 26. Piece Type 26 (Narikei2) |
| *11. Piece Type 11 (Kinsho1)* | 27. Piece Type 27 (Narikyo1) |
| *12. Piece Type 12 (Kinsho2)* | 28. Piece Type 28 (Narikyo2) |
| 21. Number of pieces on position | 29. Number of pieces on position |
| 22. Is Player 1 the current mover? | 30. Is Player 1 the current mover? |
| 23. Is Player 2 the current mover? | 31. Is Player 2 the current mover? |
| 24. Does position exist in container 0 (Board)? | 32. Does position exist in container 0 (Board)? |
| 25. Does position exist in container 1 (Hand1)? | 33. Does position exist in container 1 (Hand1)? |
| 26. Does position exist in container 2 (Hand2)? | 34. Does position exist in container 2 (Hand2)? |
| 27. Last move's from-position | 35. Last move's from-position |
| 28. Last move's to-position | 36. Last move's to-position |
| 29. Second-to-last move's from-position | 37. Second-to-last move's from-position |
| 30. Second-to-last move's to-position | 38. Second-to-last move's to-position |

## E  Details on Experimental Setup

For all training runs for transfer learning experiments, the following command-line arguments were supplied to the `train` command of Polygames (Cazenave et al., 2020):

- `--num_game 2`: Affects the number of threads used to run games per self-play client process.

- `--epoch_len 256`: Number of training batches per epoch.

- `--batchsize 128`: Batch size for model training.

- `--sync_period 32`: Affects how often models are synced.

- `--num_rollouts 400`: Number of MCTS iterations per move during self-play training.

- `--replay_capacity 100000`: Capacity of replay buffer.

- `--replay_warmup 9000`: Minimum size of replay buffer before training starts.

- `--model_name "ResConvConvLogitPoolModelV2"`: Type of architecture to use (a fully convolutional architecture with global pooling).

- `--bn`: Use of batch normalization (Ioffe & Szegedy, 2015).

- `--nnsize 2`: A value of 2 means that hidden convolutional layers each have twice as many channels as the number of channels for the state input tensors.

- `--nb_layers_per_net 6`: Number of convolutional layers per residual block.

- `--nb_nets 10`: Number of residual blocks.

- `--tournament_mode=true`: Use the tournament mode of Polygames to select checkpoints to play against in self-play.

- `--bsfinder_max_bs=800`: Upper bound on number of neural network queries batched together during inference (we used a lower value of 400 to reduce memory usage in *Breakthrough*, *Hasami Shogi*, *Kyoto Shogi*, *Minishogi*, *Shogi*, and *Tobi Shogi*).

All evaluation games in transfer learning experiments were run using the following command-line arguments for the `eval` command of Polygames:

- `--num_actor_eval=1`: Number of threads running simultaneously for a single MCTS search for the agent being evaluated.

- `--num_rollouts_eval=800`: Number of MCTS iterations per move for the agent being evaluated.

- `--num_actor_opponent=1`: Number of threads running simultaneously for a single MCTS search for the baseline agent.

- `--num_rollouts_opponent=800`: Number of MCTS iterations per move for the baseline agent.

Any parameters not listed were left at their defaults in the Polygames implementation.

# F   Details on Games and Game Variants

This section provides additional details on all the games and variants of games used throughout all the experiments described in the paper. A game with name `GameName` is selected in Polygames by providing `--game_name="GameName"` as command-line argument. For games implemented in Ludii, non-default variants are loaded by providing `--game_options "X" "Y" "Z"` as additional command-line arguments, where `X`, `Y`, and `Z` refer to one or more option strings.

## F.1   Polygames Games

For the evaluation of fully convolutional architectures, we used games as implemented directly in Polygames. Table 2 lists the exact game names used. Note that all versions of *Havannah* and *Hex* included use of the pie rule (or swap rule).

Table 2: Game implementations from Polygames used for evaluation of fully convolutional architectures. The right column shows the names used in command-line arguments.

| Game | Game Name Argument |
|---|---|
| Connect6 | `Connect6` |
| Havannah (5×5) | `Havannah5pie` |
| Havannah (8×8) | `Havannah8pie` |
| Havannah (10×10) | `Havannah10pie` |
| Hex (11×11) | `Hex11pie` |
| Hex (13×13) | `Hex13pie` |
| Hex (19×19) | `Hex19pie` |
| Minishogi | `Minishogi` |

### F.2 Ludii Game Variants

For the transfer learning experiments between variants of games, we used nine games—each with multiple variants—as implemented in Ludii: *Breakthrough*, *Broken Line*, *Diagonal Hex*, *Gomoku*, *Hex*, *HeXentafl*, *Konane*, *Pentalath*, and *Yavalath*. For each of these games, Tables 3-11 provide additional details. In each of these tables, the final column lists the number of trainable parameters in the Deep Neural Network (DNN) that is constructed for each game variant, using hyperparameters as described in Appendix E. Figure 6 provides an illustration of the two different board shapes used for *Pentalath*.

Table 3: Details on *Breakthrough* variants. This implementation of Breakthrough is loaded in Polygames using "LudiiBreakthrough.lud" as game name. By default, Breakthrough is played on an 8×8 square board.

| Variant | Options | Description | Num. Params DNN |
|---|---|---|---|
| Square6 | `"Board Size/6x6" "Board/Square"` | 6×6 square board | 188,296 |
| Square8 | `"Board Size/8x8" "Board/Square"` | 8×8 square board | 188,296 |
| Square10 | `"Board Size/10x10" "Board/Square"` | 10×10 square board | 188,296 |
| Hexagon4 | `"Board Size/4x4" "Board/Hexagon"` | 4×4 hexagonal board | 188,296 |
| Hexagon6 | `"Board Size/6x6" "Board/Hexagon"` | 6×6 hexagonal board | 188,296 |
| Hexagon8 | `"Board Size/8x8" "Board/Hexagon"` | 8×8 hexagonal board | 188,296 |

Table 4: Details on *Broken Line* variants. This implementation of Broken Line is loaded in Polygames using "LudiiBroken Line.lud" as game name.

| Variant | Options | Description | Num. Params DNN |
|---|---|---|---|
| LineSize3Hex | `"Line Size/3" "Board Size/5x5" "Board/hex"` | 5×5 hexagonal board, lines of 3 win | 222,464 |
| LineSize4Hex | `"Line Size/4" "Board Size/5x5" "Board/hex"` | 5×5 hexagonal board, lines of 4 win | 222,464 |
| LineSize5Square | `"Line Size/5" "Board Size/9x9" "Board/Square"` | 9×9 square board, lines of 5 win | 222,464 |
| LineSize6Square | `"Line Size/6" "Board Size/9x9" "Board/Square"` | 9×9 square board, lines of 6 win | 222,464 |

Table 5: Details on *Diagonal Hex* variants. This implementation of Diagonal Hex is loaded in Polygames using "LudiiDiagonal Hex.lud" as game name.

| Variant | Options | Description | Num. Params DNN |
|---|---|---|---|
| 7×7 | "Board Size/7x7" | 7×7 hexagonal board | 222,464 |
| 9×9 | "Board Size/9x9" | 9×9 hexagonal board | 222,464 |
| 11×11 | "Board Size/11x11" | 11×11 square board | 222,464 |
| 13×13 | "Board Size/13x13" | 13×13 square board | 222,464 |
| 19×19 | "Board Size/19x19" | 19×19 square board | 222,464 |

Table 6: Details on *Gomoku* variants. This implementation of Gomoku is loaded in Polygames using "Ludi-iGomoku.lud" as game name. By default, Gomoku is played on a 15×15 board.

| Variant | Options | Description | Num. Params DNN |
|---|---|---|---|
| 9×9 | "Board Size/9x9" | 9×9 square board | 180,472 |
| 13×13 | "Board Size/13x13" | 13×13 square board | 180,472 |
| 15×15 | "Board Size/15x15" | 15×15 square board | 180,472 |
| 19×19 | "Board Size/19x19" | 19×19 square board | 180,472 |

### F.3 Ludii Line Completion Games

For the evaluation of transfer between different line completion games, we used six different line completion games: *Connect6*, *Dai Hasami Shogi*, *Gomoku*, *Pentalath*, *Squava*, and *Yavalath*. Several properties of these games are listed in Table 12.

### F.4 Ludii Shogi Games

For the evaluation of transfer between different Shogi games, we used four games: *Hasami Shogi*, *Kyoto Shogi*, *Minishogi*, and *Shogi*. Several properties of these games are listed in Table 13.

### F.5 Broken Line and Diagonal Hex

*Broken Line* and *Diagonal Hex* are variations on line completion games, and *Hex*, respectively, which only take into consideration diagonal connections for the line completion and connection win conditions. On hexagonal grids, two cells are considered to be "diagonally connected" if there exists an edge that connects exactly one vertex of each of the cells. Figure 7 depicts examples of winning game states for the red player in Broken Line on a square board, Broken Line on a hexagonal board, and Diagonal Hex.

## G   Detailed Results – Zero-shot Transfer Between Game Variants

Tables 14-22 provide detailed results for all evaluations of zero-shot transfer between variants within each out of nine different games.

## H   Detailed Results – Transfer Between Game Variants With Fine-tuning

Figure 8 depicts scatterplots, with win-percentages of transferred models after finetuning against models trained on the target domain directly, for four groups of source-target pairings: (a) transfer from larger to smaller boards, (b) transfer from smaller to larger boards, (c) transfer between different board shapes altogether, and (d) trasnfer between game variants with different win (or loss) conditions. The *x*-axes on these plots provide information on the different number of training epochs used between source and target domains, as in Figure 2. Tables 23-31 provide detailed results for all evaluations of transfer performance after fine-tuning, for transfer between variants within each out of nine different games. Models are trained for 20

Table 7: Details on *Hex* variants. This implementation of Hex is loaded in Polygames using "LudiiHex.lud" as game name. By default, Hex is played on an 11×11 board.

| Variant | Options | Description | Num. Params DNN |
|---|---|---|---|
| 7×7 | `"Board Size/7x7"` | 7×7 board, standard win condition | 222,464 |
| 9×9 | `"Board Size/9x9"` | 9×9 board, standard win condition | 222,464 |
| 11×11 | `"Board Size/11x11"` | 11×11 board, standard win condition | 222,464 |
| 13×13 | `"Board Size/13x13"` | 13×13 board, standard win condition | 222,464 |
| 19×19 | `"Board Size/19x19"` | 19×19 board, standard win condition | 222,464 |
| 11×11 Misere | `"Board Size/11x11"` `"End Rules/Misere"` | 11×11 board, inverted win condition | 222,464 |

Table 8: Details on *HeXentafl* variants. This implementation of HeXentafl is loaded in Polygames using "LudiiHeXentafl.lud" as game name. By default, HeXentafl is played on a 4×4 board.

| Variant | Options | Description | Num. Params DNN |
|---|---|---|---|
| 4×4 | `"Board Size/4x4"` | 4×4 hexagonal board | 231,152 |
| 5×5 | `"Board Size/5x5"` | 5×5 hexagonal board | 231,152 |

hours on the source domain, followed by 20 hours on the target domain, and evaluated against models trained for 20 hours only on the target domain. Tables 32-40 provide additional results for a similar evaluation where we reinitialised all the parameters of the final convolutional layers before policy and value heads prior to fine-tuning. The basic idea behind this experiment was that it would lead to a more random, less biased policy generating experience from self-play at the start of a fine-tuning process, and hence may improve fine-tuning transfer in cases where full transfer produces a poor initial policy. Overall we did not observe many major changes in transfer performance.

## I   Detailed Results – Zero-shot Transfer Between Games

Tables 41-44 provide detailed results for zero-shot transfer evaluations, where source domains are different games from target domains (not just different variants).

## J   Detailed Results – Transfer Between Games With Fine-tuning

Tables 45-48 provide detailed results for evaluations of transfer performance after fine-tuning, where source domains are different games from target domains (not just different variants). Note that in these cases, the two models that play against each other do not always have exactly the same number of trainable parameters. For hidden convolutional layers, we always use twice as many channels as the number of channels in a game's state tensor representation, and this is not modified when transferring to a new domain. This means that if a source domain has a greater number of channels in its state tensor representation than the target domain, the transferred model will also still use more channels in its hidden convolutional layers than the baseline model, and vice versa when the source domain has fewer state channels. Tables 49-52 provide additional results where we adjust the number of channels of hidden convolutional layers when transferring models, prior to fine-tuning, for a more "fair" evaluation in terms of network size.

Table 9: Details on *Konane* variants. This implementation of Konane is loaded in Polygames using "Ludi-iKonane.lud" as game name. By default, Konane is played on an 8×8 board.

| Variant | Options | Description | Num. Params DNN |
|---|---|---|---|
| 6×6 | "Board Size/6x6" | 6×6 square board | 188,296 |
| 8×8 | "Board Size/8x8" | 8×8 square board | 188,296 |
| 10×10 | "Board Size/10x10" | 10×10 square board | 188,296 |
| 12×12 | "Board Size/12x12" | 12×12 square board | 188,296 |

Table 10: Details on *Pentalath* variants. This implementation of Pentalath is loaded in Polygames using "LudiiPentalath.lud" as game name. By default, Pentalath is played on half a hexagonal board.

| Variant | Options | Description | # Params DNN |
|---|---|---|---|
| HexHexBoard | "Board/HexHexBoard" | Hexagonal board | 180,472 |
| HalfHexHexBoard | "Board/HalfHexHexBoard" | Half hexagonal board | 180,472 |

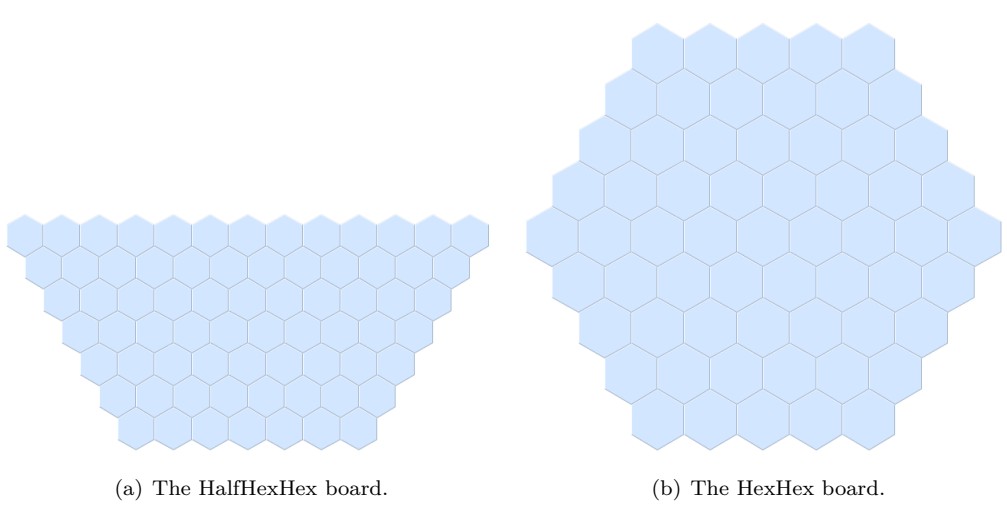

(a) The HalfHexHex board.   (b) The HexHex board.

Figure 6: Two different boards used for *Pentalath*.

Table 11: Details on *Yavalath* variants. This implementation of Yavalath is loaded in Polygames using "LudiiYavalath.lud" as game name. By default, Yavalath is played on a 5×5 board.

| Variant | Options | Description | Num. Params DNN |
|---|---|---|---|
| 3×3 | "Board Size/3x3" | 3×3 hexagonal board | 222,464 |
| 4×4 | "Board Size/4x4" | 4×4 hexagonal board | 222,464 |
| 5×5 | "Board Size/5x5" | 5×5 hexagonal board | 222,464 |
| 6×6 | "Board Size/6x6" | 6×6 hexagonal board | 222,464 |
| 7×7 | "Board Size/7x7" | 7×7 hexagonal board | 222,464 |
| 8×8 | "Board Size/8x8" | 8×8 hexagonal board | 222,464 |

Table 12: Details on different line completion games.

|  | Connect6 | Dai Hasami Shogi | Gomoku | Pentalath | Squava | Yavalath |
|---|---|---|---|---|---|---|
| Board Shape | Square | Square | Square | Hexagonal | Square | Hexagonal |
| Board Size | 19×19 | 9×9 | 9×9 | 5×5 | 5×5 | 5×5 |
| Win Line Length | 6 | 5 | 5 | 5 | 4 | 4 |
| Loss Line Length | - | - | - | - | 3 | 3 |
| Max Win Line Length | - | - | 5 | - | - | - |
| Can Move Pieces? | × | ✓ | × | × | × | × |
| Can Capture Pieces? | × | ✓ | × | ✓ | × | × |
| Uses Swap Rule? | × | × | × | × | ✓ | ✓ |
| Moves per Turn | 2* | 1 | 1 | 1 | 1 | 1 |
| State Tensor Shape | $(9, 19, 19)$ | $(9, 9, 9)$ | $(9, 9, 9)$ | $(9, 9, 17)$ | $(10, 5, 5)$ | $(10, 9, 17)$ |
| Policy Tensor Shape | $(3, 19, 19)$ | $(51, 9, 9)$ | $(3, 9, 9)$ | $(3, 9, 17)$ | $(3, 5, 5)$ | $(3, 9, 17)$ |
| Num. Params DNN | 180,472 | 188,296 | 180,472 | 180,472 | 222,464 | 222,464 |

*The first turn in Connect6 consists of only 1 move.

Table 13: Details on different Shogi games.

|  | Hasami Shogi | Kyoto Shogi | Minishogi | Shogi |
|---|---|---|---|---|
| Board Size | 9×9 | 5×5 | 5×5 | 9×9 |
| Num. Piece Types per Player | 1 | 9 | 10 | 14 |
| Can Drop Captured Pieces? | × | ✓ | ✓ | ✓ |
| State Tensor Shape | $(9, 9, 9)$ | $(28, 8, 5)$ | $(30, 8, 5)$ | $(38, 12, 9)$ |
| Policy Tensor Shape | $(51, 9, 9)$ | $(51, 8, 5)$ | $(51, 8, 5)$ | $(51, 12, 9)$ |
| Num. Params DNN | 188,296 | 1,752,908 | 2,009,752 | 3,212,648 |

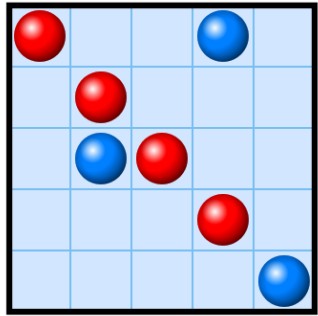
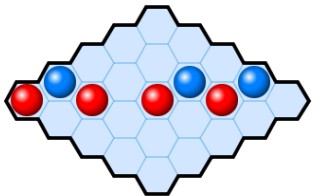
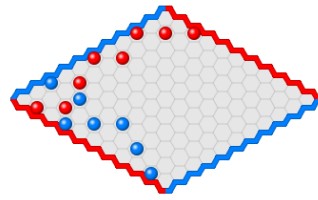

(a) A diagonal line of 4 on the square board is a win for the red player in *Broken Line*.

(b) A "diagonal" line of 4 on the hexagonal board is a win for the red player in *Broken Line*.

(c) A chain of "diagonally" connected pieces on the hexagonal board is a win for the red player in *Diagonal Hex*.

Figure 7: Examples of winning game states for the red player in *Broken Line* (on a square and hexagonal board), and *Diagonal Hex*. In both examples for *Broken Line*, the target line length was set to 4.

Table 14: Win percentage of MCTS with final checkpoint from source domain against MCTS with final checkpoint trained in target domain, evaluated in target domain (**zero-shot** transfer). Source and target domains are **different boards** in **Breakthrough**.

| Game: Breakthrough | Target Domain | | | | | |
|---|---|---|---|---|---|---|
| Source Domain | Square6 | Square8 | Square10 | Hexagon4 | Hexagon6 | Hexagon8 |
| Square6 | - | 0.00% | 7.33% | 0.00% | 0.00% | 0.67% |
| Square8 | 10.00% | - | 77.00% | 2.67% | 0.00% | 1.00% |
| Square10 | 1.33% | 0.33% | - | 0.67% | 0.00% | 0.33% |
| Hexagon4 | 0.00% | 0.00% | 0.00% | - | 0.33% | 1.33% |
| Hexagon6 | 0.67% | 0.00% | 0.00% | 12.67% | - | 39.67% |
| Hexagon8 | 0.00% | 0.00% | 0.00% | 4.00% | 5.00% | - |

Table 15: Win percentage of MCTS with final checkpoint from source domain against MCTS with final checkpoint trained in target domain, evaluated in target domain (**zero-shot** transfer). Source and target domains are **different boards** in **Broken Line**.

| Game: Broken Line | Target Domain | | | |
|---|---|---|---|---|
| Source Domain | LineSize3Hex | LineSize4Hex | LineSize5Square | LineSize6Square |
| LineSize3Hex | - | 5.67% | 0.00% | 0.00% |
| LineSize4Hex | 19.33% | - | 0.00% | 0.17% |
| LineSize5Square | 7.00% | 0.00% | - | 49.67% |
| LineSize6Square | 3.67% | 0.00% | 47.17% | - |

Table 16: Win percentage of MCTS with final checkpoint from source domain against MCTS with final checkpoint trained in target domain, evaluated in target domain (**zero-shot** transfer). Source and target domains are **different boards** in **Diagonal Hex**.

| Game: Diagonal Hex | Target Domain | | | | |
|---|---|---|---|---|---|
| Source Domain | 7×7 | 9×9 | 11×11 | 13×13 | 19×19 |
| 7×7 | - | 38.00% | 22.50% | 100.00% | 99.67% |
| 9×9 | 45.17% | - | 83.00% | 100.00% | 100.00% |
| 11×11 | 13.00% | 18.00% | - | 100.00% | 100.00% |
| 13×13 | 0.00% | 0.00% | 0.00% | - | 44.83% |
| 19×19 | 0.00% | 0.00% | 0.00% | 33.83% | - |

Table 17: Win percentage of MCTS with final checkpoint from source domain against MCTS with final checkpoint trained in target domain, evaluated in target domain (**zero-shot** transfer). Source and target domains are **different boards** in **Gomoku**.

| Game: Gomoku | Target Domain | | | |
|---|---|---|---|---|
| Source Domain | 9×9 | 13×13 | 15×15 | 19×19 |
| 9×9 | - | 44.00% | 31.67% | 18.67% |
| 13×13 | 28.17% | - | 51.33% | 62.00% |
| 15×15 | 25.50% | 40.50% | - | 66.00% |
| 19×19 | 19.83% | 32.67% | 35.67% | - |

Table 18: Win percentage of MCTS with final checkpoint from source domain against MCTS with final checkpoint trained in target domain, evaluated in target domain (**zero-shot** transfer). Source and target domains are **different variants** of **Hex**.

| Game: Hex | Target Domain | | | | | |
|---|---|---|---|---|---|---|
| Source Domain | 7×7 | 9×9 | 11×11 | 13×13 | 19×19 | 11×11 Misere |
| 7×7 | - | 38.33% | 14.67% | 76.67% | 91.67% | 0.00% |
| 9×9 | 21.67% | - | 56.33% | 100.00% | 100.00% | 0.00% |
| 11×11 | 20.33% | 30.33% | - | 100.00% | 100.00% | 0.00% |
| 13×13 | 4.67% | 0.67% | 0.00% | - | 100.00% | 0.00% |
| 19×19 | 0.00% | 0.00% | 0.00% | 0.00% | - | 0.00% |
| 11×11 Misere | 0.00% | 0.00% | 0.00% | 0.00% | 0.00% | - |

Table 19: Win percentage of MCTS with final checkpoint from source domain against MCTS with final checkpoint trained in target domain, evaluated in target domain (**zero-shot** transfer). Source and target domains are **different boards** in **HeXentafl**.

| Game: HeXentafl | Target Domain | |
|---|---|---|
| Source Domain | 4×4 | 5×5 |
| 4×4 | - | 15.50% |
| 5×5 | 9.67% | - |

Table 20: Win percentage of MCTS with final checkpoint from source domain against MCTS with final checkpoint trained in target domain, evaluated in target domain (**zero-shot** transfer). Source and target domains are **different boards** in **Konane**.

| Game: Konane | Target Domain | | | |
|---|---|---|---|---|
| Source Domain | 6×6 | 8×8 | 10×10 | 12×12 |
| 6×6 | - | 3.00% | 14.67% | 63.33% |
| 8×8 | 31.00% | - | 94.00% | 100.00% |
| 10×10 | 12.00% | 3.33% | - | 99.67% |
| 12×12 | 8.00% | 0.00% | 2.00% | - |

Table 21: Win percentage of MCTS with final checkpoint from source domain against MCTS with final checkpoint trained in target domain, evaluated in target domain (**zero-shot** transfer). Source and target domains are **different boards** in **Pentalath**.

| Game: Pentalath | Target Domain | |
|---|---|---|
| Source Domain | HexHexBoard | HalfHexHexBoard |
| HexHexBoard | - | 26.67% |
| HalfHexHexBoard | 18.00% | - |

Table 22: Win percentage of MCTS with final checkpoint from source domain against MCTS with final checkpoint trained in target domain, evaluated in target domain (**zero-shot** transfer). Source and target domains are **different boards** in **Yavalath**.

| Game: Yavalath | Target Domain | | | | | |
|---|---|---|---|---|---|---|
| Source Domain | 3×3 | 4×4 | 5×5 | 6×6 | 7×7 | 8×8 |
| 3×3 | - | 10.83% | 4.33% | 1.67% | 0.67% | 0.33% |
| 4×4 | 29.83% | - | 29.83% | 15.33% | 7.67% | 4.33% |
| 5×5 | 10.33% | 12.17% | - | 30.33% | 34.00% | 25.00% |
| 6×6 | 8.17% | 20.17% | 41.17% | - | 45.33% | 58.00% |
| 7×7 | 8.50% | 21.00% | 33.00% | 38.00% | - | 53.33% |
| 8×8 | 7.67% | 13.00% | 31.00% | 29.83% | 47.67% | - |

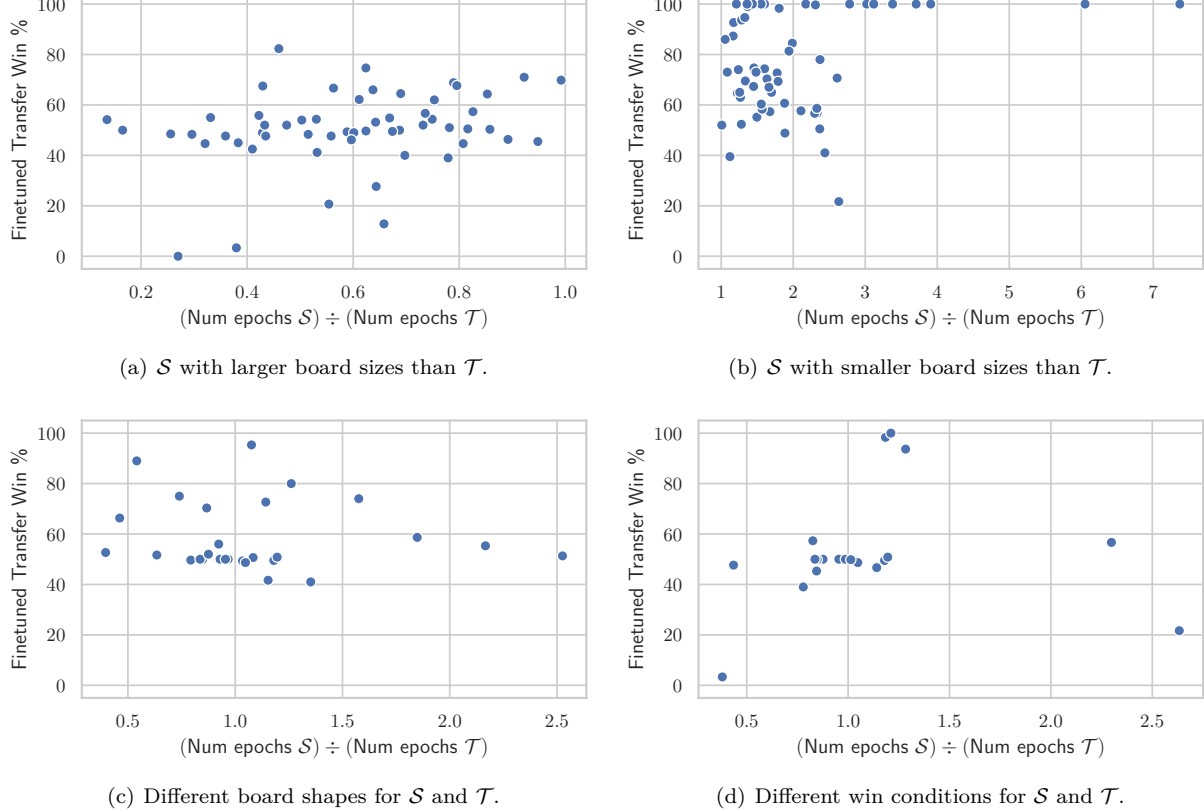

(a) $\mathcal{S}$ with larger board sizes than $\mathcal{T}$.

(b) $\mathcal{S}$ with smaller board sizes than $\mathcal{T}$.

(c) Different board shapes for $\mathcal{S}$ and $\mathcal{T}$.

(d) Different win conditions for $\mathcal{S}$ and $\mathcal{T}$.

Figure 8: Win percentages of models trained on $\mathcal{S}$ and subsequently fine-tuned on $\mathcal{T}$, against models trained only on $\mathcal{T}$—evaluated on $\mathcal{T}$. Results above 50% may be interpreted as beneficial transfer—assuming that pre-training on $\mathcal{S}$ is "for free", without depleting any budget—whereas results below 50% indicate negative transfer.

Table 23: Win percentage of MCTS with final checkpoint from source domain against MCTS with final checkpoint trained in target domain, evaluated in target domain after **fine-tuning**. Source and target domains are **different boards** in **Breakthrough**.

| Game: Breakthrough | Target Domain | | | | | |
|---|---|---|---|---|---|---|
| Source Domain | Square6 | Square8 | Square10 | Hexagon4 | Hexagon6 | Hexagon8 |
| Square6 | - | 87.33% | 99.00% | 50.67% | 74.00% | 51.33% |
| Square8 | 50.33% | - | 92.67% | 50.00% | 41.00% | 55.33% |
| Square10 | 52.00% | 64.33% | - | 49.67% | 41.67% | 58.67% |
| Hexagon4 | 56.00% | 95.33% | 80.00% | - | 74.67% | 56.67% |
| Hexagon6 | 51.67% | 75.00% | 70.33% | 50.00% | - | 74.33% |
| Hexagon8 | 52.67% | 66.33% | 89.00% | 49.00% | 74.67% | - |

Table 24: Win percentage of MCTS with final checkpoint from source domain against MCTS with final checkpoint trained in target domain, evaluated in target domain after **fine-tuning**. Source and target domains are **different boards** in **Broken Line**.

| Game: Broken Line | Target Domain | | | |
|---|---|---|---|---|
| Source Domain | LineSize3Hex | LineSize4Hex | LineSize5Square | LineSize6Square |
| LineSize3Hex | - | 46.67% | 50.00% | 50.00% |
| LineSize4Hex | 50.00% | - | 49.83% | 50.00% |
| LineSize5Square | 49.33% | 49.50% | - | 50.00% |
| LineSize6Square | 48.67% | 50.83% | 49.83% | - |

Table 25: Win percentage of MCTS with final checkpoint from source domain against MCTS with final checkpoint trained in target domain, evaluated in target domain after **fine-tuning**. Source and target domains are **different boards** in **Diagonal Hex**.

| Game: Diagonal Hex | Target Domain | | | | |
|---|---|---|---|---|---|
| Source Domain | 7×7 | 9×9 | 11×11 | 13×13 | 19×19 |
| 7×7 | - | 48.83% | 84.50% | 100.00% | 100.00% |
| 9×9 | 54.33% | - | 86.00% | 100.00% | 100.00% |
| 11×11 | 54.00% | 45.50% | - | 100.00% | 100.00% |
| 13×13 | 55.00% | 49.67% | 12.83% | - | 41.00% |
| 19×19 | 54.17% | 48.50% | 0.00% | 42.50% | - |

Table 26: Win percentage of MCTS with final checkpoint from source domain against MCTS with final checkpoint trained in target domain, evaluated in target domain after **fine-tuning**. Source and target domains are **different boards** in **Gomoku**.

| Game: Gomoku | Target Domain | | | |
|---|---|---|---|---|
| Source Domain | 9×9 | 13×13 | 15×15 | 19×19 |
| 9×9 | - | 69.50% | 70.33% | 78.00% |
| 13×13 | 54.33% | - | 64.67% | 72.67% |
| 15×15 | 62.17% | 50.50% | - | 67.33% |
| 19×19 | 55.50% | 66.67% | 64.50% | - |

Table 27: Win percentage of MCTS with final checkpoint from source domain against MCTS with final checkpoint trained in target domain, evaluated in target domain after **fine-tuning**. Source and target domains are **different variants** of **Hex**.

| Game: Hex | Target Domain | | | | | |
|---|---|---|---|---|---|---|
| Source Domain | 7×7 | 9×9 | 11×11 | 13×13 | 19×19 | 11×11 Misere |
| 7×7 | - | 69.33% | 81.33% | 100.00% | 100.00% | 56.67% |
| 9×9 | 47.67% | - | 73.00% | 100.00% | 100.00% | 93.67% |
| 11×11 | 48.33% | 71.00% | - | 100.00% | 100.00% | 98.33% |
| 13×13 | 47.67% | 27.67% | 40.00% | - | 100.00% | 57.33% |
| 19×19 | 50.00% | 48.33% | 44.67% | 82.33% | - | 3.33% |
| 11×11 Misere | 47.67% | 39.00% | 45.33% | 100.00% | 21.67% | - |

Table 28: Win percentage of MCTS with final checkpoint from source domain against MCTS with final checkpoint trained in target domain, evaluated in target domain after **fine-tuning**. Source and target domains are **different boards** in **HeXentafl**.

| Game: HeXentafl | Target Domain | |
|---|---|---|
| Source Domain | 4×4 | 5×5 |
| 4×4 | - | 50.50% |
| 5×5 | 55.83% | - |

Table 29: Win percentage of MCTS with final checkpoint from source domain against MCTS with final checkpoint trained in target domain, evaluated in target domain after **fine-tuning**. Source and target domains are **different boards** in **Konane**.

| Game: Konane | Target Domain | | | |
|---|---|---|---|---|
| Source Domain | 6×6 | 8×8 | 10×10 | 12×12 |
| 6×6 | - | 52.33% | 65.00% | 99.67% |
| 8×8 | 51.00% | - | 94.67% | 98.33% |
| 10×10 | 49.33% | 62.00% | - | 100.00% |
| 12×12 | 52.00% | 20.67% | 56.67% | - |

Table 30: Win percentage of MCTS with final checkpoint from source domain against MCTS with final checkpoint trained in target domain, evaluated in target domain after **fine-tuning**. Source and target domains are **different boards** in **Pentalath**.

| Game: Pentalath | Target Domain | |
|---|---|---|
| Source Domain | HexHexBoard | HalfHexHexBoard |
| HexHexBoard | - | 72.67% |
| HalfHexHexBoard | 52.00% | - |

Table 31: Win percentage of MCTS with final checkpoint from source domain against MCTS with final checkpoint trained in target domain, evaluated in target domain after **fine-tuning**. Source and target domains are **different boards** in **Yavalath**.

| Game: Yavalath | Target Domain | | | | | |
|---|---|---|---|---|---|---|
| Source Domain | 3×3 | 4×4 | 5×5 | 6×6 | 7×7 | 8×8 |
| 3×3 | - | 39.50% | 67.00% | 57.33% | 57.67% | 70.67% |
| 4×4 | 46.33% | - | 73.00% | 55.17% | 60.67% | 58.67% |
| 5×5 | 49.00% | 49.50% | - | 52.00% | 63.00% | 58.33% |
| 6×6 | 46.17% | 54.83% | 69.83% | - | 65.00% | 60.33% |
| 7×7 | 52.00% | 41.17% | 68.83% | 67.67% | - | 74.00% |
| 8×8 | 45.00% | 67.50% | 66.00% | 53.17% | 44.67% | - |

Table 32: Win percentage of MCTS with final checkpoint from source domain against MCTS with final checkpoint trained in target domain, evaluated in target domain after **fine-tuning** with **reinitialised** final layers. Source and target domains are **different boards** in **Breakthrough**.

| Game: Breakthrough | Target Domain | | | | | |
|---|---|---|---|---|---|---|
| Source Domain | Square6 | Square8 | Square10 | Hexagon4 | Hexagon6 | Hexagon8 |
| Square6 | - | 92.67% | 96.33% | 48.33% | 66.67% | 65.33% |
| Square8 | 57.00% | - | 88.33% | 49.67% | 60.33% | 65.33% |
| Square10 | 52.33% | 53.33% | - | 49.33% | 42.33% | 38.00% |
| Hexagon4 | 47.67% | 77.67% | 95.00% | - | 84.33% | 73.00% |
| Hexagon6 | 53.00% | 86.67% | 68.67% | 49.67% | - | 74.00% |
| Hexagon8 | 52.33% | 66.00% | 93.33% | 52.00% | 74.00% | - |

Table 33: Win percentage of MCTS with final checkpoint from source domain against MCTS with final checkpoint trained in target domain, evaluated in target domain after **fine-tuning** with **reinitialised** final layers. Source and target domains are **different boards** in **Broken Line**.

| Game: Broken Line | Target Domain | | | |
|---|---|---|---|---|
| Source Domain | LineSize3Hex | LineSize4Hex | LineSize5Square | LineSize6Square |
| LineSize3Hex | - | 49.00% | 50.00% | 50.00% |
| LineSize4Hex | 49.00% | - | 50.00% | 49.83% |
| LineSize5Square | 50.00% | 50.50% | - | 50.00% |
| LineSize6Square | 49.67% | 49.67% | 49.67% | - |

Table 34: Win percentage of MCTS with final checkpoint from source domain against MCTS with final checkpoint trained in target domain, evaluated in target domain after **fine-tuning** with **reinitialised** final layers. Source and target domains are **different boards** in **Diagonal Hex**.

| Game: Diagonal Hex | Target Domain | | | | |
|---|---|---|---|---|---|
| Source Domain | 7×7 | 9×9 | 11×11 | 13×13 | 19×19 |
| 7×7 | - | 51.00% | 86.67% | 100.00% | 100.00% |
| 9×9 | 50.33% | - | 89.67% | 100.00% | 100.00% |
| 11×11 | 51.33% | 46.83% | - | 100.00% | 100.00% |
| 13×13 | 54.83% | 49.67% | 15.00% | - | 53.00% |
| 19×19 | 52.83% | 49.33% | 6.50% | 45.17% | - |

Table 35: Win percentage of MCTS with final checkpoint from source domain against MCTS with final checkpoint trained in target domain, evaluated in target domain after **fine-tuning** with **reinitialised** final layers. Source and target domains are **different boards** in **Gomoku**.

| Game: Gomoku | Target Domain | | | |
|---|---|---|---|---|
| Source Domain | 9×9 | 13×13 | 15×15 | 19×19 |
| 9×9 | - | 68.00% | 65.33% | 70.00% |
| 13×13 | 61.83% | - | 65.33% | 74.00% |
| 15×15 | 59.33% | 58.33% | - | 71.67% |
| 19×19 | 55.33% | 55.17% | 57.17% | - |

Table 36: Win percentage of MCTS with final checkpoint from source domain against MCTS with final checkpoint trained in target domain, evaluated in target domain after **fine-tuning** with **reinitialised** final layers. Source and target domains are **different variants** of **Hex**.

| Game: Hex | Target Domain | | | | | |
|---|---|---|---|---|---|---|
| Source Domain | 7×7 | 9×9 | 11×11 | 13×13 | 19×19 | 11×11 Misere |
| 7×7 | - | 68.00% | 74.00% | 100.00% | 100.00% | 87.33% |
| 9×9 | 49.00% | - | 72.67% | 99.67% | 100.00% | 96.33% |
| 11×11 | 49.00% | 50.33% | - | 100.00% | 100.00% | 93.00% |
| 13×13 | 51.33% | 41.33% | 40.00% | - | 100.00% | 73.33% |
| 19×19 | 49.67% | 43.00% | 36.00% | 99.00% | - | 0.67% |
| 11×11 Misere | 47.00% | 37.67% | 41.00% | 100.00% | 84.00% | - |

Table 37: Win percentage of MCTS with final checkpoint from source domain against MCTS with final checkpoint trained in target domain, evaluated in target domain after **fine-tuning** with **reinitialised** final layers. Source and target domains are **different boards** in **HeXentafl**.

| Game: HeXentafl | Target Domain | |
|---|---|---|
| Source Domain | 4×4 | 5×5 |
| 4×4 | - | 52.17% |
| 5×5 | 43.17% | - |

Table 38: Win percentage of MCTS with final checkpoint from source domain against MCTS with final checkpoint trained in target domain, evaluated in target domain after **fine-tuning** with **reinitialised** final layers. Source and target domains are **different boards** in **Konane**.

| Game: Konane | Target Domain | | | |
|---|---|---|---|---|
| Source Domain | 6×6 | 8×8 | 10×10 | 12×12 |
| 6×6 | - | 54.00% | 76.33% | 98.33% |
| 8×8 | 51.67% | - | 95.67% | 99.33% |
| 10×10 | 50.67% | 36.00% | - | 99.00% |
| 12×12 | 51.33% | 14.67% | 38.00% | - |

Table 39: Win percentage of MCTS with final checkpoint from source domain against MCTS with final checkpoint trained in target domain, evaluated in target domain after **fine-tuning** with **reinitialised** final layers. Source and target domains are **different boards** in **Pentalath**.

| Game: Pentalath | Target Domain | |
|---|---|---|
| Source Domain | HexHexBoard | HalfHexHexBoard |
| HexHexBoard | - | 65.67% |
| HalfHexHexBoard | 51.67% | - |

Table 40: Win percentage of MCTS with final checkpoint from source domain against MCTS with final checkpoint trained in target domain, evaluated in target domain after **fine-tuning** with **reinitialised** final layers. Source and target domains are **different boards** in **Yavalath**.

| Game: Yavalath | Target Domain | | | | | |
|---|---|---|---|---|---|---|
| Source Domain | 3×3 | 4×4 | 5×5 | 6×6 | 7×7 | 8×8 |
| 3×3 | - | 50.50% | 56.17% | 65.67% | 72.00% | 70.67% |
| 4×4 | 48.33% | - | 69.00% | 68.50% | 63.00% | 47.33% |
| 5×5 | 50.33% | 51.67% | - | 44.00% | 57.67% | 44.33% |
| 6×6 | 53.17% | 60.50% | 68.17% | - | 68.33% | 53.67% |
| 7×7 | 49.33% | 54.83% | 68.83% | 56.67% | - | 59.00% |
| 8×8 | 51.17% | 43.33% | 57.67% | 45.83% | 68.00% | - |

Table 41: Win percentage of MCTS with final checkpoint from source domain against MCTS with final checkpoint trained in target domain, evaluated in target domain (**zero-shot** transfer). Source and target domains are **different line-completion games**.

| | Target Domain | | | | | |
|---|---|---|---|---|---|---|
| Source Domain | Connect6 | Dai Hasami Shogi | Gomoku | Pentalath | Squava | Yavalath |
| Connect6 | - | 0.00% | 2.33% | 0.00% | 1.00% | 0.33% |
| Dai Hasami Shogi | 0.67% | - | 1.33% | 0.00% | 0.67% | 1.67% |
| Gomoku | 36.67% | 0.00% | - | 0.33% | 2.67% | 1.33% |
| Pentalath | 11.67% | 0.00% | 4.33% | - | 2.00% | 1.33% |
| Squava | 16.00% | 0.00% | 0.33% | 0.00% | - | 2.00% |
| Yavalath | 0.00% | 0.00% | 0.00% | 0.33% | 1.67% | - |

Table 42: Win percentage of MCTS with final checkpoint from source domain against MCTS with final checkpoint trained in target domain, evaluated in target domain (**zero-shot** transfer). Source and target domains are **different Shogi games**.

| | Target Domain | | | |
|---|---|---|---|---|
| Source Domain | Hasami Shogi | Kyoto Shogi | Minishogi | Shogi |
| Hasami Shogi | - | 1.33% | 0.33% | 52.67% |
| Kyoto Shogi | 39.83% | - | 3.00% | 44.67% |
| Minishogi | 47.17% | 16.17% | - | 97.00% |
| Shogi | 23.83% | 1.67% | 0.00% | - |

Table 43: Win percentage of MCTS with final checkpoint from ***Broken Line* variants** against MCTS with final checkpoint trained in target domain, evaluated in target domain (**zero-shot** transfer). Target domains are **different line completion games**.

| Source (Broken Line) | Connect6 | Dai Hasami Shogi | Gomoku | Pentalath | Squava | Yavalath |
|---|---|---|---|---|---|---|
| | | Target Domain | | | | |
| LineSize3Hex | 0.00% | 0.00% | 0.00% | 0.00% | 0.67% | 1.67% |
| LineSize4Hex | 0.00% | 0.00% | 0.00% | 0.00% | 0.33% | 1.67% |
| LineSize5Square | 31.33% | 0.00% | 1.00% | 0.33% | 0.67% | 1.33% |
| LineSize6Square | 32.00% | 0.00% | 1.00% | 1.67% | 0.33% | 2.00% |

Table 44: Win percentage of MCTS with final checkpoint from ***Diagonal Hex* variants** against MCTS with final checkpoint trained in target domain, evaluated in target domain (**zero-shot** transfer). Target domains are **different variants of *Hex*.**

| Source (Diagonal Hex) | 7×7 | 9×9 | 11×11 | 11×11 Misere | 13×13 | 19×19 |
|---|---|---|---|---|---|---|
| | | | Target (Hex) | | | |
| 7×7 | 0.00% | 0.00% | 0.00% | 0.00% | 0.00% | 10.33% |
| 9×9 | 0.00% | 0.00% | 0.00% | 0.00% | 0.00% | 15.67% |
| 11×11 | 0.00% | 0.00% | 0.00% | 0.00% | 0.00% | 28.33% |
| 13×13 | 0.00% | 0.00% | 0.00% | 0.00% | 0.00% | 9.00% |
| 19×19 | 0.00% | 0.00% | 0.00% | 0.00% | 0.00% | 24.33% |

Table 45: Win percentage of MCTS with final checkpoint from source domain against MCTS with final checkpoint trained in target domain, evaluated in target domain (after **fine-tuning**). Source and target domains are **different line-completion games**.

| Source Domain | Connect6 | Dai Hasami Shogi | Gomoku | Pentalath | Squava | Yavalath |
|---|---|---|---|---|---|---|
| | | Target Domain | | | | |
| Connect6 | - | 54.00% | 53.17% | 58.33% | 45.50% | 63.33% |
| Dai Hasami Shogi | 54.67% | - | 54.50% | 53.67% | 48.00% | 72.33% |
| Gomoku | 95.00% | 50.17% | - | 59.33% | 48.50% | 49.67% |
| Pentalath | 92.00% | 53.33% | 57.67% | - | 46.67% | 50.50% |
| Squava | 94.33% | 51.00% | 56.67% | 64.00% | - | 75.17% |
| Yavalath | 43.00% | 53.33% | 56.00% | 56.00% | 45.00% | - |

Table 46: Win percentage of MCTS with final checkpoint from source domain against MCTS with final checkpoint trained in target domain, evaluated in target domain (after **fine-tuning**). Source and target domains are **different Shogi games**.

| Source Domain | Hasami Shogi | Kyoto Shogi | Minishogi | Shogi |
|---|---|---|---|---|
| | | Target Domain | | |
| Hasami Shogi | - | 38.17% | 40.17% | 89.00% |
| Kyoto Shogi | 45.67% | - | 35.67% | 70.00% |
| Minishogi | 52.00% | 63.17% | - | 86.67% |
| Shogi | 49.67 | 75.83% | 36.00% | - |

Table 47: Win percentage of MCTS with final checkpoint from ***Broken Line* variants** against MCTS with final checkpoint trained in target domain, evaluated in target domain (after **fine-tuning**). Target domains are **different line completion games**.

| Source (Broken Line) | Connect6 | Dai Hasami Shogi | Gomoku | Pentalath | Squava | Yavalath |
|---|---|---|---|---|---|---|
| LineSize3Hex | 56.00% | 52.00% | 54.83% | 53.67% | 49.00% | 47.17% |
| LineSize4Hex | 64.67% | 52.83% | 53.83% | 56.33% | 48.00% | 68.00% |
| LineSize5Square | 88.67% | 53.67% | 53.83% | 66.00% | 47.33% | 64.00% |
| LineSize6Square | 90.00% | 52.33% | 50.67% | 52.00% | 46.00% | 58.67% |

Table 48: Win percentage of MCTS with final checkpoint from ***Diagonal Hex* variants** against MCTS with final checkpoint trained in target domain, evaluated in target domain (after **fine-tuning**). Target domains are **different variants of Hex**.

| Source (Diagonal Hex) | 7×7 | 9×9 | 11×11 | 11×11 Misere | 13×13 | 19×19 |
|---|---|---|---|---|---|---|
| 7×7 | 51.00% | 59.00% | 15.33% | 48.67% | 99.33% | 80.00% |
| 9×9 | 51.67% | 50.33% | 19.00% | 53.33% | 100.00% | 70.67% |
| 11×11 | 46.00% | 19.67% | 6.33% | 23.00% | 97.33% | 40.67% |
| 13×13 | 47.00% | 56.67% | 5.33% | 0.67% | 0.00% | 20.00% |
| 19×19 | 45.00% | 54.67% | 28.67% | 1.67% | 0.00% | 45.33% |

Table 49: Win percentage of MCTS with final checkpoint from source domain against MCTS with final checkpoint trained in target domain, evaluated in target domain (after **fine-tuning**). Source and target domains are **different line-completion games**.

| Source Domain | Connect6 | Dai Hasami Shogi | Gomoku | Pentalath | Squava | Yavalath |
|---|---|---|---|---|---|---|
| Connect6 | - | 53.83% | 57.50% | 69.00% | 51.00% | 74.00% |
| Dai Hasami Shogi | 55.33% | - | 57.67% | 59.00% | 48.67% | 68.83% |
| Gomoku | 93.00% | 52.00% | - | 60.33% | 46.00% | 62.67% |
| Pentalath | 76.67% | 48.33% | 59.83% | - | 47.33% | 58.33% |
| Squava | 40.67% | 50.00% | 58.17% | 58.67% | - | 69.50% |
| Yavalath | 70.33% | 52.00% | 51.67% | 53.00% | 49.00% | - |

Table 50: Win percentage of MCTS with final checkpoint from source domain against MCTS with final checkpoint trained in target domain, evaluated in target domain (after **fine-tuning**, with **number of channels** in hidden convolutional layers **adjusted to be equal**). Source and target domains are **different Shogi games**.

| Source Domain | Hasami Shogi | Kyoto Shogi | Minishogi | Shogi |
|---|---|---|---|---|
| Hasami Shogi | - | 35.83% | 34.67% | 67.33% |
| Kyoto Shogi | 48.00% | - | 33.67% | 63.33% |
| Minishogi | 50.00% | 58.00% | - | 65.33% |
| Shogi | 49.67% | 45.67% | 45.67% | - |

Table 51: Win percentage of MCTS with final checkpoint from **_Broken Line_ variants** against MCTS with final checkpoint trained in target domain, evaluated in target domain (after **fine-tuning**, with **number of channels** in hidden convolutional layers **adjusted to be equal**). Target domains are **different line completion games**.

|  | Target Domain | | | | | |
|---|---|---|---|---|---|---|
| Source (Broken Line) | Connect6 | Dai Hasami Shogi | Gomoku | Pentalath | Squava | Yavalath |
| LineSize3Hex | 46.67% | 50.00% | 48.17% | 56.00% | 45.67% | 74.17% |
| LineSize4Hex | 45.67% | 54.33% | 52.17% | 60.00% | 48.33% | 66.67% |
| LineSize5Square | 94.00% | 49.33% | 54.33% | 63.00% | 48.33% | 70.00% |
| LineSize6Square | 82.67% | 49.67% | 50.17% | 47.33% | 47.00% | 72.00% |

Table 52: Win percentage of MCTS with final checkpoint from **_Diagonal Hex_ variants** against MCTS with final checkpoint trained in target domain, evaluated in target domain (after **fine-tuning**, with **number of channels** in hidden convolutional layers **adjusted to be equal**). Target domains are **different variants of Hex**.

|  | Target (Hex) | | | | | |
|---|---|---|---|---|---|---|
| Source (Diagonal Hex) | 7×7 | 9×9 | 11×11 | 11×11 Misere | 13×13 | 19×19 |
| 7×7 | 50.67% | 31.67% | 56.33% | 8.67% | 99.67% | 100.00% |
| 9×9 | 47.00% | 48.33% | 42.00% | 40.00% | 98.67% | 66.67% |
| 11×11 | 47.67% | 25.67% | 42.33% | 20.67% | 87.00% | 12.00% |
| 13×13 | 49.33% | 44.33% | 8.33% | 1.33% | 0.00% | 56.67% |
| 19×19 | 50.00% | 50.67% | 28.33% | 2.00% | 0.00% | 10.67% |

