# OpenReview forum: "Towards a General Transfer Approach for Policy-Value Networks"
_TMLR — Accepted by TMLR_

### Review · Reviewer_xv6k · 2023-09-25

**Summary Of Contributions:**

This paper presents transfer learning experiments between various board games defined in the Ludii general game system. To enable zero-shot transfer, the authors define neural network architectures for AlphaZero's state and policy networks using global pooling and convolutional output layers, respectively. The authors also define a transfer protocol to translate network input and action channels between source and target games utilizing the domain specific language (DSL) definitions of these games. The experiments show examples of strong and weak zero-shot and fine-tuned transfer performance. Of particular importance, the experiments show that stronger performance on games with large boards can often be achieved by training on variants of the same game with smaller boards.

**Audience:**

Yes

**Broader Impact Concerns:**

Broader Impact Statement not required.

**Claims And Evidence:**

Yes

**Requested Changes:**

## Critical Changes
- The lines in Figure 1 are confusing. Even ignoring the lines, it is very difficult to compare architectures from Figure 1.
- Figures 2, 4, and 5 do not do a great enough job of presenting the experiment results. They are enough to support some of the authors' conclusions, but not nearly all of them. Since the experiments are so important to this paper's contributions, the results have to be communicated more thoroughly in the main paper. This is not to say that all of the result tables should be moved into the main paper, but that the main paper's tables and figures should support more of the analysis. That may mean adding some tables or figures to the main paper, or reworking the current figures.

## Changes to Strengthen the Submission
- Figure 3 takes up a lot of space show one particular case of bilateral zero-shot transfer in Pentalath with ostensibly notable win percentages. The case for why this transfer is notable could be stronger.
- The first paragraph of Section 5 is largely repeated in other places. Its unique points could be better emphasized or portions could be cut to save space.
- One sentence in Section 4.1 states that 24 hours of computation were used to train each model but every other reference to computation said 20 hours. Please clarify.

**Strengths And Weaknesses:**

## Overview

The paper contains good, straightforward ideas. The experimental design is conceptually simple and solid. For example, global pooling and convolutional output layers are good ideas for facilitating transfer between board games. However, technical explanations, some experiment details, and the analysis could be improved.


## Technical Explanations

Section 3 is the only technical, non-background or experiment section describing how transfer is enabled between the Ludii games, but it defers a lot of information important to understand the experiments to Appendix B and C. Even after reading Appendix B and C, I found it difficult to understand all of the design choices involved in the transfer protocol. Improving the presentation of this  material and moving it into the main paper would make it easier for readers to understand the experiments and build on the insights from the results.

For an example of an important detail that I did not know until I read the supplementary material, piece types are sometimes heuristically linked by minimum tree edit distance. With this and other design choices, I wonder how often these imperfect translations were required, how large the actual semantic differences were, and if there was any correlation between the number and extent of imperfect translations with transfer performance. I do not see this information in the paper or supplementary material (a weakness of the analysis, along with others that I describe later).

Another example is that the paper states in multiple places that to formulate the transfer protocol, the authors assume that source and target games are identical games with different state or action representations. After reading the supplementary material, I think the authors mean that they started by ensuring their transfer protocol that would be lossless if the source and target games were identical and building additional facilities into the protocol to handle different games. This idea does not come through readily in the main paper alone.

The way actions are transferred between movement and placement games is an example detail that I do not understand, even after reading Section B.3.

Related to the fact that Section 3 is a relatively small part of the paper is that ideas and arguments related to using information from DSL game definitions get lost. The DSL information is being used to match input and action channels between representations across games, but is it being used to make the chance of beneficial transfer more likely? The abstract mentions that a purpose of this paper is to argue for greater adoption of benchmark problems that are described with DSLs but I do not see where this argument is made.


## Experiment Details

A limitation of the experiments to me is that it appears a single model was trained on each game. How much of the results can be attributed to initialization variance? I do not believe the initialization variance would be enough to invalidate the conclusions that come from observations across many game pairs, but it may explain individual outliers.

Is zero really a good baseline for zero-shot transfer? I am not sure. I suspect that the models are poor on large boards, judging from the results in Figure 1 and the fact that zero-shot transfer from training on small boards often appears so successful. What score do you get if you play against the trained models with newly initialized models?


## Analysis

It is difficult for me to get too much out of Figures 2, 4, and 5 when, in most cases, win % does not look like it is strongly correlated with the length of training in S. The fact that transfer from small games to larger ones tends to work better is maybe the only fact that is readily apparent to me from these figures. However, given the clusters of points in many of the plots, it looks like there might be much more illustrative game groupings and comparisons. I find the tables in the supplementary material to be more informative.

What are the relationships between zero-shot and fine-tuned performance? I think that this is an important factor in assessing the significance of the zero-shot performance results and the practical applicability of training regimes suggested by this work. If a 30% or 40% win rate can be achieved on zero-shot transfer by training on a small board, but fine tuning does not yield any further benefit on a large board, then the prospect of training on a small board and transferring to a large board is less feasible than the zero-shot results suggest.

---

> ### Author Response · Authors · 2023-10-21
> **Official Comment by Authors (Part 1/2)**
>
> Thanks for your review.
>
> We have just uploaded an initial revision with some minor improvements. In this comment, we also provide initial replies to some of your remarks. **We also aim to upload another revision with more substantial improvements as soon as possible** (currently expecting to finish this by Wednesday). We will notify you again here once this second revision is available.
>
> > Section 3 is the only technical, non-background or experiment section [...]
>
> Thanks for your specific examples of aspects that were not sufficiently clear. We will improve this in our larger revision and upload it as soon as possible.
>
> > For an example of an important detail that I did not know until I read the supplementary material [...]
>
> We will analyse the relations between these aspects and the results, and report any findings in our larger revision as soon as possible.
>
> For the games listed in Table 11 (supplementary material): Squava and Yavalath have a state channel to encode whether or not the Swap Rule has been used, which the other four games in this table don’t. This does not get heuristically translated to anything though, this is simply a channel that is completely present or absent and therefore ignored or randomly/zero-initialised accordingly. Dai Hasami Shogi is the only game out of these six that involves movement (as opposed to only placement) of pieces, so it has more channels in the policy head than the other five games.
>
> For the games listed in Table 12 (supplementary material): all four games in this table have different numbers of piece types (1, 9, 10, and 14 per player), so they all require imperfect matching between piece type channels. Many piece names are common across all four games, so many channels get matched “perfectly” based on piece names, although sometimes this is incorrect: Hasami Shogi uses the names “Fuhyo” and “Tokin” for the sole piece types owned by the two different players, whereas in the other three games, use “Fuhyo” is a piece type that both players have, and “Tokin” is a promoted version of that piece type. When transferring from a game with fewer piece types to one with more piece types, there will always be some piece types for which the “most similar” source channel is identified via tree edit distances.
>
> > Another example is that the paper states in multiple places that to formulate the transfer protocol, the authors assume that [...]
>
> That is indeed what we mean. We will clarify.
>
> > Related to the fact that Section 3 is a relatively small part of the paper is that ideas and arguments related to using information from DSL game definitions get lost. The DSL information is being used to match input and action channels between representations across games, but is it being used to make the chance of beneficial transfer more likely? The abstract mentions that a purpose of this paper is to argue for greater adoption of benchmark problems that are described with DSLs but I do not see where this argument is made.
>
> We would make both arguments separately: (1) that DSLs can be leveraged to make it more likely to get beneficial transfer (especially in the zero-shot case), and (2) that it would be valuable to look more at benchmarks where problems are described with DSLs.
>
> Our reasoning for argument (1) is that, if a problem is not described in some form that can be provided as input to some algorithm, zero-shot transfer is simply hopeless (except if mappings are fully hardcoded by a programmer). The same is true for humans: if a human has played the game of Hex for a while, and we suddenly start playing Misere Hex against that human (i.e., we flip around the win condition), but we don’t inform the human that the rules have changed (don’t provide a description of the problem), we cannot expect the human to do anything other than play as if we were still playing with the original win condition, and the human will have terrible zero-shot performance. Zero-shot transfer without providing descriptions of problems as input is similarly hopeless in the general case for any machine learning approach, and DSLs are a convenient and powerful way to provide information about the problem as input.
>
> (2) is more of a philosophical argument: if we want the general public (not just machine learning experts / programmers) to be able to describe the problems that they want AI to tackle for them, it is more convenient for them if they are able to do so in a DSL than in a general-purpose programming language (natural language would be even more convenient, but is also probably more challenging on the AI side).

---

> ### Author Response · Authors · 2023-10-21
> **Official Comment by Authors (Part 2/2)**
>
> > A limitation of the experiments to me is that it appears a single model was trained on each game. How much of the results can be attributed to initialization variance? I do not believe the initialization variance would be enough to invalidate the conclusions that come from observations across many game pairs, but it may explain individual outliers.
>
> While we cannot completely rule out the possibility that noise might explain individual outliers, we have some reasons to believe this is unlikely. AlphaZero-like training has been demonstrated to generally be stable and produce similar learning curves over different random seeds in other work, see e.g. Figure S3 in the supplementary material of the original AlphaZero paper (Silver et al., 2018). Hence, we also expect this of our initial models (ones that are not transferred or zero-shot-transferred). As for fine-tuned models: in Subsection 4.4, we report two variations (new fine-tuning runs, with different random seeds on top of other changes), and we find little variety in results. So, these results have been confirmed to be stable across three different random seeds (+ additional actual changes).
>
> > Is zero really a good baseline for zero-shot transfer? I am not sure. I suspect that the models are poor on large boards, judging from the results in Figure 1 and the fact that zero-shot transfer from training on small boards often appears so successful. What score do you get if you play against the trained models with newly initialized models?
>
> For many cases, we are sure that the trained models are substantially stronger than randomly initialised models. We used the same hyperparameters, hardware and time as reported by (Soemers et al., 2022) for their (non-transfer-learning) experiments, and they showed trained MCTSes with substantially fewer iterations convincingly outperforming untrained MCTSes with many more iterations, for a set of games that partially overlaps with our set of games. Indeed, some of the biggest board sizes that we considered were not evaluated by them though. So, it is possible that the non-transferred models trained directly on the biggest board sizes are weak. But this does not mean that the baseline is wrong. This simply means that there may be a risk of getting weak performance when training on very big boards, and that training on smaller boards + zero-shot transfer is an effective way to mitigate that risk.
>
> > ## Analysis
>
> We are looking into a more detailed analysis for our larger revision, to be uploaded as soon as possible.
>
> > The lines in Figure 1 are confusing. Even ignoring the lines, it is very difficult to compare architectures from Figure 1.
>
> We have updated Figure 1.
>
> > Figures 2/4/5, Figure 3
>
> We will come back to these points and specify our changes once we have finalised our full revision in a few days.
>
> > One sentence in Section 4.1 states that 24 hours of computation were used to train each model but every other reference to computation said 20 hours. Please clarify.
>
> The experiment described in Section 4.1 (training and evaluation of many different architectures, no transfer learning here) used 24 hours (and used games implemented directly in C++, in Polygames). All other experiments (new trainings, all used for transfer learning and as baselines against transferred models, with games implemented in Ludii’s game description language rather than C++) used 20 hours.

---

> ### Author Response · Authors · 2023-10-25
>
> This is to notify you that we have just uploaded a more substantial revision.
>
> All changes are summarised close to the top of this page, under "Changes Since Last Submission". Here, we will summarise again the ones that were made specifically in response to your comments (and not already addressed in our previous reply above):
>
> - Clarified and edited some text in the first few paragraphs of Section 3, to make the procedures used for mapping channels and transferring parameters more clear. Added Subsections 3.1 and 3.2 as completely new subsections for Section 3 (both largely based on what used to be only in an appendix: the appendix is still there too, with some more detail).
> - Explicitly mentioned the example of how channels are sometimes heuristically mapped, as is the case for Piece Type channels (based on tree edit distances) in Section 3.
> - Briefly discussed the number of models trained per source game, and implications in terms of statistical reliability of results, in Section 4.
> - The figure with four scatterplots for four different subsets of finetuning results, with ratios of training epochs between source and target domain on the $x$-axis, has been moved into an Appendix (this is now Figure 8). As you correctly pointed out, there was not much of an observable relationship between the $x$- and $y$-axes. Instead, the same space is in the main paper is now occupied by a new figure (now Figure 3), which has the zero-shot transfer playing strength on the $x$-axis. This lets us make some more observations about the extent to which negative transfer occurs (if at all) during finetuning, or only during the initial transfer. The discussion of results in Subsection 4.2 has also been extended for this.
> - Moved the figure illustrating the two different board shapes used for *Pentalath* (used to be Figure 3) down into the Appendices
>
> Thanks again for your comments, and please let us know if there are any concerns that remain unaddressed, or new concerns.

---

### Review · Reviewer_wV6n · 2023-09-26

**Summary Of Contributions:**

This paper studies the problem of transferring a policy-value network from an alpha-go architecture between zero-sum games with different state and action spaces.  The study is broadly empirical focused on the performance of a "simple baseline" approach, which consists of 1) fully convolutional-pooling networks, with no final fully connected layers, allowing for architecture that can be re-run at various widths and 2) a standardization of the action and state descriptions which allows for a canonical way of re-mapping channels.  The study found some surprising cases of transfer, including policies training on smaller boards and transferring to larger board sizes working better than policies trained for the same amount of time (but few epochs), on the larger boards directly.

**Audience:**

Yes

**Broader Impact Concerns:**

I do not see any specific ethics concerns which would warrant a specific Broader Impact Statement.

**Claims And Evidence:**

Yes

**Requested Changes:**

[critical change] It should be clarified how the transfer works across games with different channels clearly enough that someone could reproduce the results of the paper

[Non-critical changes]
* it should be mentioned somewhere in the abstract that this study is of alpha-go-esq policies.  I had incorrectly assumed it would be policy gradient algorithms.
* Plasticity loss should be mentioned somewhere. Specifically:
    * At the bottom of page 9 where it mentions resetting the last layer, this is also done in the plasticity loss literature
    * At the bottom of page 8 plasticity loss should probably be listed as a possible cause of negative transfer.
* it should be mentioned prior work has trained policies to work on several board sizes at once, for instance Kata-go.

**Strengths And Weaknesses:**

**Strengths**
The strengths of this paper are that it is very deliberate in its experimental design, and is careful to be accurate in its claims. For the most part the results are presented in a clear way, and I feel like I have learned something interesting from the paper.

**Weaknesses**
The main weakness I see with the paper is that the way the state and action channels are handled during transfer is unclear, and pretty critical to understanding the experimental design. I would suggest walking through a specific example of the transfer being done.  The description of this in section 3 is very high-level and it is difficult to understand what this means in terms of code or work-flow, which parts the programmer does and which parts are done automatically.

To try to clarify the confusion, the example of this used in the paper about the empty board causes more questions than answers, since it is unclear as to how it can be automatically determined which channels of the state space correspond to "empty"

The other example in this section is written in a way I find very hard to parse:
"For example, a plane that encodes the presence of friendly pieces is semantically equivalent regardless of the game in which it encodes the data in terms of the raw data that it encodes, ...."
It is also unclear in this example what is meant by "semantically equivalent" and how that can be automatically determined.

Further down where it mentions that channels can be copied, reordered, duplicated or removed, it's not clear how any of those operations are done, or how that might vary from game to game. As it stands, I could not reproduce the transfer protocol because of this.

---

> ### Author Response · Authors · 2023-10-21
>
> Thanks for your review.
>
> We have just uploaded an initial revision with some minor improvements. In this comment, we also provide initial replies to some of your remarks. **We also aim to upload another revision with more substantial improvements as soon as possible** (currently expecting to finish this by Wednesday). We will notify you again here once this second revision is available.
>
> > [critical change] It should be clarified how the transfer works across games with different channels clearly enough that someone could reproduce the results of the paper
>
> We will come back to this point and specify our changes once we have finalised our full revision in a few days. But the basic idea is as follows. When different games are described in a standardised DSL, such as Ludii’s game description language, this also translates to a standardised and consistent internal state representation. From a raw data perspective, the notion of “empty” means the same across all games: it means that there is nothing (no piece) in a certain position. So, a channel that distinguishes between emty/non-empty can be readily transferred between any pair of games. Identifying which channel this is in each game is trivial, since each game simply has one channel that is explicitly defined as such. Figuring out which channels these are is no more complex than it is to identify which channel in a program such as AlphaGo or AlphaZero is the channel encoding for empty positions, or which three channels are encoding for Red/Blue/Green in typical image-processing networks.
>
> Of course, it is possible that whether or not a position is empty has different implications in terms of different game’s rules. For example, empty positions might be legal destinations for moves in some game A, and they might be illegal destinations for moves in some other game B. But we largely ignore such subtleties in our work: we simply transfer based on the raw data encoded by different channels.
>
> > it should be mentioned somewhere in the abstract that this study is of alpha-go-esq policies. I had incorrectly assumed it would be policy gradient algorithms.
>
> We have updated the final sentence (now split up into the final two sentences) of the abstract to clarify this.
>
> > Plasticity loss should be mentioned somewhere. Specifically:
> >   - At the bottom of page 9 where it mentions resetting the last layer, this is also done in the plasticity loss literature
> >   - At the bottom of page 8 plasticity loss should probably be listed as a possible cause of negative transfer.
>
> We have mentioned this in the two specified locations, with appropriate new references.
>
> > it should be mentioned prior work has trained policies to work on several board sizes at once, for instance Kata-go.
>
> We have added one more citation to (Wu, 2019) to Subsection 2.3, to make this more explicit (in addition to other places where we already cited this work).

---

> ### Author Response · Authors · 2023-10-25
>
> This is to notify you that we have just uploaded a more substantial revision.
>
> All changes are summarised close to the top of this page, under "Changes Since Last Submission". Here, we will summarise again the ones that were made specifically in response to your comments (and not already addressed in our previous reply above):
>
> - Clarified and edited some text in the first few paragraphs of Section 3, to make the procedures used for mapping channels and transferring parameters more clear. Added Subsections 3.1 and 3.2 as completely new subsections for Section 3 (both largely based on what used to be only in an appendix: the appendix is still there too, with some more detail).
> - Added a new appendix and table (in this revision: Appendix D and Table 1) explaining, as an example, how channels are mapped between the games of *Minishogi* and *Shogi*.
>
> Thanks again for your comments, and please let us know if there are any concerns that remain unaddressed, or new concerns.

---

### Review · Reviewer_nhxA · 2023-10-18

**Summary Of Contributions:**

The paper describes and empirically evaluates an approach for transfer learning in two-player AlphaZero-style game-playing agents, as applied in the Ludii game-playing system.

The transfer-learning approach allows a policy-value network to be reused between game variants and disparate games, possibly having radically different state- and action spaces. This generalizes upon previous work in this area, which mainly focuses on just state transfers or transfers between similar games only.  Another contribution is a demonstration of transfer learning on a wide array of two-player games (much more extensive than seen before).

**Audience:**

Yes

**Broader Impact Concerns:**

None.

**Claims And Evidence:**

Yes

**Requested Changes:**

As stated above, the paper is mostly well-written and structured.  However, it will benefit from a more detailed technical discussion of the method in Section 3 (in the main text as opposed to in an appendix). This suggestion is not critical for acceptance but will result in added readability, as one should be able to grasp all main ideas without having to refer (too much) to the appendices.
Apart from that, there are only a few minor remarks, as listed below.

Page 4:
„Following Subsection 2.1, ...“  may be clearer phrased as
„As in Subsection 2.1, ...“

„... DNNs ... if it only has different values for one ...“ should be
„... DNNs ... if they have different values for only one ...“


Page 5:
„This section discusses experiments ...“
„This section discusses the experiments ...“
„80 iterations per move“ and „400 iterations per move“.
Is it intentional to talk about „iterations“ as opposed to (the more common) „simulations per move“? (as the term simulation is sometimes also used to refer to the MCST playout/rollout phase.)

Page 6:
The text in Figure 1 is somewhat „fuzzy“.

Page 9:
„... in 4.4.“ should be „... in Subsection 4.4“
„... throughout 4.2 and 4.3“ should be „... throughout Subsections 4.2 and 4.3“

Page 10:
Figure 5: the x- and y-axis label font is too large --- looks odd.

**Strengths And Weaknesses:**

The main idea is to have the network's tensor channels encode the internal state variables of the Ludii game-playing system, which are mostly common among all games. The main benefit of such an approach is that transfer learning can be applied more generally, for example, for disparate state- and action-spaces. But it also has drawbacks (acknowledged in the paper), like requiring having only convolutional layers in the network (i.e., not linear layers towards the end) and, arguably, somewhat suboptimal (for learning) representation of the game states.

The approach seems technically sound, albeit it comes across (arguably) as a bit specific for the game-playing system at hand, i.e., represented games must have a similar internal implementation; although, as the authors argue in their conclusion, this is likely to be the case for all such systems using a DSL. In any case, overall, the approach is nonetheless of interest and offers something to be learned by other researchers in their field.

The approach's effectiveness is thoroughly evaluated by transferring network parameters between multiple games in the Ludii game-playing system, both slight variants in board shape/sizes and between unlike games. The playing strength of the resulting agents is thoroughly evaluated against baseline agents, both with and without further training.  As expected, the effectiveness of the transfer widely varies between games. However, some common themes emerge, for example, that transferring from smaller to larger game variants may be beneficial. The extensive and thorough evaluation is one of the strong points of the paper.

As for the presentation, the paper is mostly well-written and structured (see some remarks below). Related work is adequately acknowledged.

---

> ### Author Response · Authors · 2023-10-21
>
> Thanks for your review.
>
> We have just uploaded an initial revision with some minor improvements. In this comment, we also provide initial replies to some of your remarks. **We also aim to upload another revision with more substantial improvements as soon as possible** (currently expecting to finish this by Wednesday). We will notify you again here once this second revision is available.
>
> > As stated above, the paper is mostly well-written and structured. However, it will benefit from a more detailed technical discussion of the method in Section 3 (in the main text as opposed to in an appendix). This suggestion is not critical for acceptance but will result in added readability, as one should be able to grasp all main ideas without having to refer (too much) to the appendices.
>
> We will add more details in the main text as you suggest, and notify you here again once we have uploaded our revision.
>
> > Is it intentional to talk about „iterations“ as opposed to (the more common) „simulations per move“? (as the term simulation is sometimes also used to refer to the MCST playout/rollout phase.)
>
> We think that the term “iterations” is more precise. Indeed, the term “simulation” is often associated with solely the play-out/rollout part of MCTS, whereas we interpret “iteration” as the full sequence of Selection+Playout+Expansion+Backpropagation. In the case of MCTS agents with trained networks, the play-out phase is cut out altogether, so there is not much of a simulation left to speak of (except if we consider tree traversal to be a simulation).
>
> Furthermore, the untrained MCTS agents (which are used as baselines in the experiment of Subsection 4.1) in Polygames are actually implemented to use 10 random rollouts, rather than just 1, within each iteration. So, these agents use 800 iterations per move (800 tree traversals and backpropagation), but 8000 random rollouts (every iteration backs up an outcome averaged from 10 random rollouts). This also makes iteration a more precise term than simulation.
>
> We notice now that, while such subtle implementation details of the untrained MCTS are visible in the Polygames source code (which we used as a framework), these are not yet explicitly listed in our paper. We will add one more Appendix to do so in our larger revision in a few days.
>
> > other remarks on phrasing
>
> All your other remarks on improvements to phrasing have been applied.

---

> ### Author Response · Authors · 2023-10-25
>
> This is to notify you that we have just uploaded a more substantial revision.
>
> All changes are summarised close to the top of this page, under "Changes Since Last Submission". Here, we will summarise again the ones that were made specifically in response to your comments (and not already addressed in our previous reply above):
>
> - Clarified and edited some text in the first few paragraphs of Section 3, to make the procedures used for mapping channels and transferring parameters more clear. Added Subsections 3.1 and 3.2 as completely new subsections for Section 3 (both largely based on what used to be only in an appendix: the appendix is still there too, with some more detail).
> - Clarified in Subsection 4.1 that the untrained UCTs back up average outcomes of 10 random rollouts per iteration of MCTS.
>
> Thanks again for your comments, and please let us know if there are any concerns that remain unaddressed, or new concerns.

---

### Comment · Reviewer_nhxA · 2023-11-09
**Revisions**

I'v read through the revised transcript and I am content with the changes and believe they adequately address the reservations we had.

---

### Decision · Action_Editor_VcNb · 2023-11-20

**Recommendation:** Accept as is

**Comment:**

The authors addressed the primary concerns and weaknesses raised by the reviewers.  There was agreement that moving some of the material from the appendix, both giving more evidence for the paper's claims and clarity in the approach.

All the reviewers agreed that the paper provides sufficient evidence for its claims, and it makes a contribution for which a portion of TMLR's audience would be eager to understand, and possibly build from.

**Audience:**

Transferring knowledge (zero-shot or few-shot) between disparate environments has a broad audience.  This particular approach uses knowledge inherent in the rules of the game expressed as a DSL.  This should be of interest to a considerable portion of TMLR's audience, even if it may be specific to the specific specification language used.

**Claims And Evidence:**

Claims are empirical and backed up by thorough experimentation.